# 🧱 Block Transformer: Global-to-Local Language Modeling for Fast Inference

**Namgyu Ho**[1,2†*]    **Sangmin Bae**[1*]    **Taehyeon Kim**[1]    **Hyunjik Jo**[2]    **Yireun Kim**[2]
**Tal Schuster**[3]    **Adam Fisch**[3]    **James Thorne**[1‡]    **Se-Young Yun**[1‡]
[1]KAIST AI    [2]LG AI Research    [3]Google DeepMind
{itsnamgyu, bsmn0223, thorne, yunseyoung}@kaist.ac.kr
https://github.com/itsnamgyu/block-transformer

## Abstract

We introduce the Block Transformer which adopts hierarchical global-to-local modeling to autoregressive transformers to mitigate the inference bottlenecks associated with self-attention. Self-attention requires the key-value (KV) cache of all previous sequences to be retrieved from memory at every decoding step to retrieve context information, leading to two primary bottlenecks during batch inference. First, there is a significant delay in obtaining the first token, as the information of the *entire* prompt must first be processed to prefill the KV cache. Second, computation of subsequent tokens is bottlenecked by the high memory I/O demand of fetching the entire KV cache, which grows linearly with sequence length, incurring quadratic memory reads overall. We design the Block Transformer to strategically mitigate these costs, by incorporating coarsity and locality into an integrated global-to-local architecture. At the lower layers, we aggregate tokens into fixed size *blocks* to apply attention across the entire sequence at coarse-grained detail, to capture the global context while minimizing KV cache overhead. At upper layers, we apply attention within each block to decode individual tokens, to model fine-grained details with a lightweight local KV cache. We pretrain vanilla and Block Transformers from scratch and demonstrate that Block Transformers reach 10–20x inference throughput compared to vanilla transformers with equivalent perplexity and zero-shot task performance.

## 1 Introduction

Generating tokens with transformer-based autoregressive language models (LMs) is costly due to the self-attention mechanism that attends to all preceding tokens [6, 77]. To minimize computation, it is common to cache the key-value (KV) states of all tokens during decoding. However, significant initial latency remains from processing the KV states of all prompt tokens during the first decoding step. Also, while subsequent decoding steps only need to compute a single token, this is often bottlenecked by the memory access required to retrieve the KV states of all previous tokens. While numerous techniques have been proposed to reduce attention costs [23, 42, 2], there has been limited research on effective architectures that structurally mitigate attention overheads in transformer-based LMs.

Hierarchical architectures [59, 36, 22, 56, 54, 57, 28, 50, 84] have shown potential in efficiently modeling long sequences such as *character*-level text or pixel-level images by pooling inputs into coarser units. While most of these employ upsampling to regress back to a finer level for attention computation, several approaches [50, 84] further enhance efficiency through local processing within each pooled unit. Nevertheless, they have not recognized or explored the potential of applying local processing to alleviate the overheads of autoregressive inference, explained above.

---

[†]Work done during an internship at LG AI Research.    [*]Equal contribution.    [‡]Corresponding authors.

38th Conference on Neural Information Processing Systems (NeurIPS 2024).

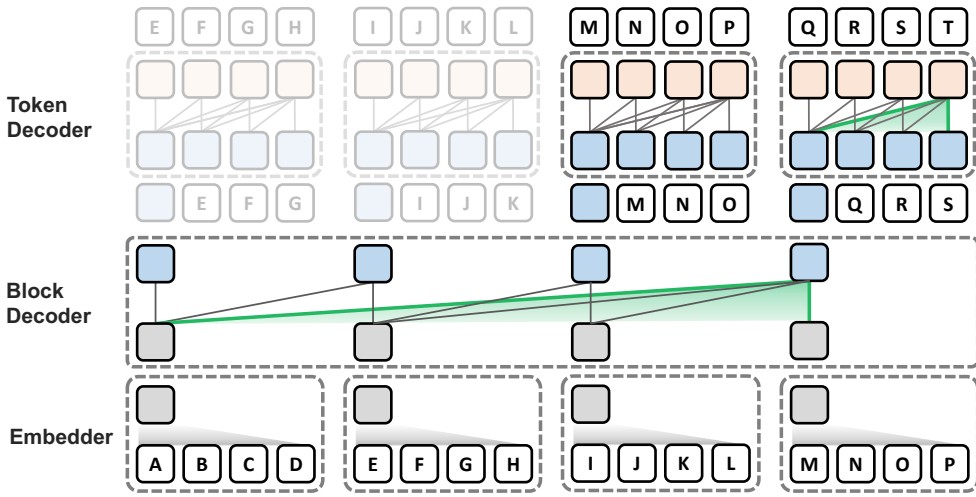

Figure 1: An overview of the Block Transformer architecture with a block length of $L_B = 4$. Each letter represents a standard subword token. Attention is restricted within the dotted boundaries in the decoders. The KV values outside the boundary of the currently decoded block do not need to be retrieved during decoding, and can be discarded from memory. Consequently, given the prompt tokens, A, B, ..., L, the shaded parts do not need to be computed in the token decoder.

In this paper, we present the Block Transformer architecture which employs a global-to-local hierarchical structure at the *subword*-level to holistically maximize the *inference* benefits of global-to-local processing for self-attention. We begin with a brief summary of our architecture as shown in Figure 1:

The lower layers are designed to efficiently capture the global input context by applying attention at the coarse level. First, the lightweight **embedder** maps each block of $L_B$ subword tokens into a single *input block embedding*. These are passed to the **block decoder**, an autoregressive transformer that applies self-attention between blocks to output a *output block embedding*, or *context embedding*, used by the upper layers to decode the next block. The upper layers are designed to predict the next block in finer detail, returning to the level of individual tokens. Here, the **token decoder** autoregressively decodes the tokens in the next block. Relying on the context embedding from the lower layers for global context, fine-level attention is applied only within the current block of $L_B$ tokens, using a minuscule local KV cache.

This structure alleviates a wide range of key inference bottlenecks with only simple modifications to the standard transformer. At the lower layers, coarse processing reduces the amount of KV cache storage by a factor of $L_B$ and memory access by $L_B{}^2$. At the upper layers, attention is applied within the local block of $L_B$ tokens, nearly eliminating its overhead. Consider a prompt with $L$ tokens. While vanilla transformer layers would have to prefill and store the KV cache of $L$ tokens and retrieve these at every decoding step, the upper layers of the Block Transformer only need to see up to $L_B \ll L$ most recent tokens, nearly eliminating prefill computation and KV cache overheads.

While prior work [84] has introduced similar structures, it has largely overlooked their potential benefits in autoregressive inference. Notably, they focus on optimizing overall FLOPs for efficient *pretraining* by exploiting lightweight local modules that simply map between coarse and fine representations. Our approach challenges this viewpoint, uncovering vital roles of both the *global* block decoder and *local* token decoder in language modeling. Ablations reveal that a more balanced parameter allocation across the global and local modules enhances performance *and* inference throughput, attributed to significant reduction in self-attention overheads within the local module.

Extensive experiments on models up to 1.4 billion parameters show that Block Transformers notably improve inference throughput for both prefill- and decode-intensive scenarios, achieving **10–20×** **gains in throughput** compared to vanilla transformers with equivalent perplexity or zero-shot task performance. Despite the architectural restriction of global-to-local attention, our models show a comparable ability to utilize full context on recent long-context benchmarks, such as PG19 [62] and Needle-In-a-Haystack [39], when compared to their vanilla transformer counterparts. In addition, we show that it is possible to uptrain pretrained vanilla models into Block Transformers, closely approaching the performance of those pretrained from scratch, using minimal training budget.

## 2 Block Transformer

The Block Transformer employs global and local attention mechanisms with hierarchical paradigm by separating the comprehension of the full context and detailed interactions into two distinct stages. Specifically, global context is captured at lower layers in coarse block-level granularity, where each block consists of a fixed number of tokens aggregated into a single embedding. The local dependencies are resolved at upper layers, where multiple subword tokens are decoded in an autoregressive manner using the context embedding from the block decoder, with local attention.

The Block Transformer consists of three components:

1. **Embedder**: The embedder aggregates each block of $L_B$ tokens into an input block embedding.
2. **Block decoder**: The block decoder applies self-attention across the full sequence of blocks to model global dependencies.
3. **Token decoder**: The token decoder applies self-attention within each block to handle fine-grained local dependencies and decode individual tokens.

We outline the design and list efficiency benefits for each component in the following subsections. To lay the groundwork for detailed cost analysis, we begin with a simplified primer on key bottlenecks in autoregressive transformers, assuming a single accelerator device.

### 2.1 A primer on the key bottlenecks of autoregressive transformers

**Key costs of autoregressive transformers**   These can be categorized into *compute*, *parameter memory* and *KV cache memory*. (1) Compute refers to arithmetic operations, dominated by matrix multiplications in the attention and feedforward layers. The number of floating-point operations (FLOPs) is proportional to the number of non-embedding parameters and total tokens, i.e., sequence length × batch size. (2) Parameters must be fully stored in device memory and retrieved at *each* forward or backward pass, regardless of the sequence length and batch size. (3) During inference, the KV cache of previously computed sequences are typically cached in memory, and retrieved at *each* forward pass, i.e., decoding step. Similar to compute, its size is proportional to sequence length × batch size.

**Case study of Llama 7B [75]**   The compute cost of *each* token is roughly $7 \times 2 = 14$GFLOPs, where 2 comes from the multiply-accumulate operation used in matrix-vector multiplication [40]. The memory size of the model parameters is $7 \times 2 = 14$GB under 16-bit, or 2-byte, floating-point precision. The size of the KV cache of a *single* token is 512KB. While this seems minuscule compared to the parameters, the total KV cache can quickly outsize the parameters with more tokens; for sequence length 2048 and batch size 16, the KV cache occupies 16GB. To translate this to *wall-clock time*, note that the compute throughput (FLOP/s) of accelerator devices is 2–3 orders of magnitude higher than HBM memory bandwidth (bytes/s)—a gap which is widening exponentially [33].

**Computation scenarios for autoregressive LMs**   These can be broadly divided into *training* and inference. During training, every token is processed in parallel. For a precise cost analysis of inference, we further dissect it into the *prefill stage* and the *decode stage*. To generate a response to a given prompt, all input tokens must first be processed to obtain and cache their KV values in each layer, allowing subsequent tokens to access them for self-attention. This is referred to as the prefill stage. Next is the decode stage, where subsequent tokens are generated one at a time. In each forward step, only one token per batch sample is computed, but the KV cache of *all* preceding tokens must be loaded from memory.

**Primary bottlenecks per computation scenario vary significantly**   (1) During training, all tokens are processed in parallel, thus compute demands far outweigh parameter memory access, which remains constant. (2) Similarly, the prefill stage is typically compute-bound, as all input tokens are processed in parallel. (3) In contrast, the decode stage, given sufficient sequence length, is heavily memory-bound, as only one token per batch sample is computed in each forward pass, while parameter and KV cache memory access demands are significant. Specifically, KV cache memory access exceeds parameter memory access when sequence length and batch size are large, while parameter memory dominates when they are small. A larger batch size helps reduce the relative cost of parameter memory access by amortizing it over batch samples. Our proposed Block Transformer design optimizes inference, especially in batch decoding, where KV cache impacts throughput.

## 2.2 Embedder

For the embedder, we prioritize simplicity, given the small block lengths $L_B = 1, 2, 4, 8$ considered in our study. Our primary design uses a lookup table $E_{\text{emb}} \in \mathbb{R}^{V \times D_{\text{emb}}}$ to retrieve and concatenate trainable token embeddings. We set the embedding dimension to $D_{\text{emb}} = D/L_B$, where $D$ is the primary model dimension, used throughout the block and token decoders. We also consider variants which incorporate small encoder transformers (Appendix F), but these do not yield performance improvements (Section 3.4).

## 2.3 Block decoder

The block decoder aims to contextualize block representations by attending to preceding blocks, utilizing the embedder's output as input. This autoregressive transformer operates at the block level, producing output block embeddings, or *context embeddings*[2], that enable the token decoder to autoregressively decode the subsequent block's token contents. Given input block embeddings from the embedder, derived from input tokens $x_{0:i \times L_B - 1}$, the block decoder outputs a context embedding which contains the information to predict $x_{i \times L_B:(i+1) \times L_B - 1}$.

This approach mitigates the quadratic costs of self-attention by using coarse-grained block inputs instead of individual tokens, while preserving global modeling capabilities and ease of hardware acceleration of dense attention [85]. This reduces the context length of a given sequence by $L_B$ compared to a vanilla transformer. We compare the block decoder with vanilla transformer layers, in terms of the key bottlenecks discussed in Section 2.1:

- *Computation* is reduced by $L_B$ due to reduced input units.
- *Parameter memory access* is reduced by $L_B$ due to reduced decoding frequency.
- *KV cache size* is reduced by $L_B$ due to coarsity.[3] *KV cache access* is reduced by $L_B^2$ due to reduced size *and* decoding frequency.

## 2.4 Token decoder

The token decoder decodes the individual tokens of the next block, taking the form of an autoregressive transformer with a separate embedding table $E_{\text{tok}} \in \mathbb{R}^{V \times D_{\text{tok}}}$ and classifier. Each block is processed independently, relying on the context embedding as the sole source of information on previous input blocks. Only the *local* KV cache of the current block needs to be stored in memory and accessed at each decoding step. Conveniently, none of the prompt tokens in previous blocks are attended by future tokens, allowing the prefill stage to be skipped entirely–except for the most recent block.

The relative size of this local KV cache is smaller by a factor of $R = L/L_B$ when compared to the global KV cache of vanilla transformers, which spans the entire sequence length $L$. For standard lengths $L = 2048$ and $L_B = 4$, the reduction factor is a staggering $R = 256$.[4] We compare the token decoder with vanilla transformer layers, with respect to key bottlenecks:

- *Computation* is reduced to near-zero during prefill. Training computes remains the same.
- *Parameter memory access* remains the same.
- *KV cache size* is reduced by $R = L/L_B$ with $L_B \ll L$, practically eliminating its memory footprint.[3] *KV cache memory access* is equally reduced by $R$.

The key to designing the token decoder lies in how to incorporate the context embedding into the decoding process. In our main design, we project the context embedding into prefix tokens, enabling further refinement of the global context. Expanding the number of prefix tokens, i.e., *prefix length*, broadens the token decoder's computation width and allows for finer attention to context information and improved performance, similar to pause tokens [34]. Note that this is only feasible due to local processing, whereas extra tokens in vanilla transformers layers would exacerbate the overhead of KV cache. Extra computation incurred by prefix tokens has minimal effect on inference throughput as inference is largely memory-bound. While we also considered summation and cross-attention based variants (Appendix F), these proved less effective than our main method (Section 3.4).

---

[2]We use the term *context embedding*, as it is the sole source of context information for the token decoder.
[3]Reduced KV cache memory footprint can lower hardware requirements or allow for larger batch sizes, which can increase throughput by further amortizing parameter memory access.
[4]Models continue to support longer contexts, like Gemini [67] with 2M tokens, effectively making $R$ larger.

Table 1: Performance comparison between vanilla and block transformer models. For a clear comparison, we highlight an example where the vanilla and our models achieve comparable levels of training loss. We measure the perplexity of LAMBADA [58] and WikiText [49], and the accuracy of HellaSwag [86], PIQA [11], and ARC-easy [21] benchmarks. Memory refers to the amount of memory allocated per sample, measured in megabytes, while throughput is measured in units of 1K tokens per second. * refers to variants trained with random-length padding[5].

| Models | # Parameter | | | Zero-shot Eval | | | | | Memory↓ | | Throughput↑ | |
|---|---|---|---|---|---|---|---|---|---|---|---|---|
| | Total | N-Emb | Loss↓ | LD↓ | WK↓ | HS↑ | PQ↑ | ARC↑ | Prefill[h] | Decode[h] | Prefill[h] | Decode[h] |
| Vanilla | 31M | 5M | 3.002 | 282.7 | 78.4 | 26.47 | 57.97 | 37.10 | 355.0 | 38.5 | 10.8 | 41.6 |
| | 70M | 19M | 2.820 | 67.2 | 46.9 | 27.20 | 59.73 | 40.24 | 390.0 | 76.8 | 6.9 | 19.1 |
| | 160M | 85M | 2.476 | 20.2 | 28.5 | 29.80 | 64.22 | 46.85 | 675.0 | 229.6 | 2.3 | 6.2 |
| | 410M | 302M | 2.224 | 10.0 | 20.1 | 35.05 | 68.10 | 51.68 | 1140.0 | 608.2 | 0.8 | 2.1 |
| Block | 33M* | 5M | 3.578 | 2359.9 | 134.2 | 26.25 | 55.90 | 35.17 | 25.0 | 5.0 | 272.3 | 809.5 |
| | 77M* | 19M | 3.181 | 390.5 | 80.1 | 27.21 | 57.69 | 38.31 | 48.9 | 9.9 | 175.3 | 421.4 |
| | 170M* | 85M | 2.753 | 67.9 | 43.7 | 28.28 | 62.22 | 43.43 | 56.3 | 29.1 | 59.0 | 134.7 |
| | 420M | 302M | 2.445 | 29.5 | 27.7 | 31.13 | 64.35 | 48.48 | 105.0 | 77.2 | 21.0 | 44.1 |
| | 1.0B | 805M | 2.268 | 16.5 | 21.4 | 34.68 | 68.18 | 52.26 | 130.2 | 102.8 | 19.8 | 42.5 |
| | 1.4B | 1.2B | 2.188 | 12.2 | 19.1 | 36.66 | 68.63 | 54.63 | 194.2 | 153.9 | 12.4 | 25.7 |

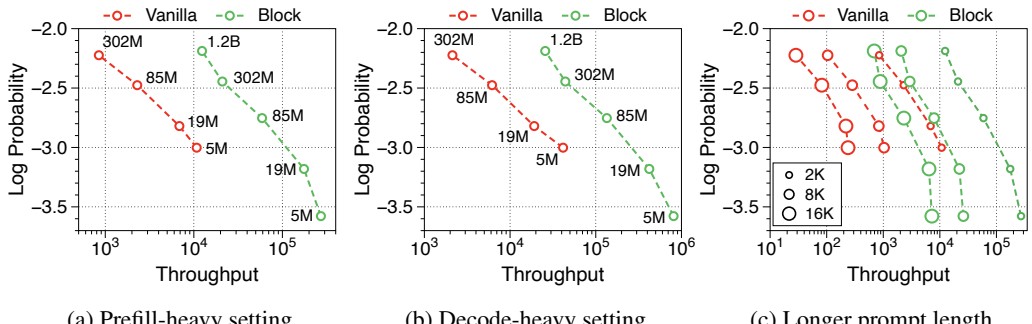

(a) Prefill-heavy setting  (b) Decode-heavy setting  (c) Longer prompt length

Figure 2: Pareto frontier of throughput to language modeling performance. Throughput denotes the number of generated tokens per second, and the numbers next to each point represent the number of non embedding parameters. (a) Pareto frontier in the prefill-heavy setting. (b) Pareto frontier in the decode-heavy setting. (c) Throughput in the prefill-heavy setting with varying prompt lengths. Each point corresponds to the same order of model sizes as in the left figures.

# 3 Experiments

## 3.1 Experimental setup

We use the transformer architecture of Pythia [10], and train both vanilla and Block Transformer models on the Pile [30, 9] with a context length of 2048. The models are pretrained on 300B tokens, which corresponds to about 1.5 epochs. We employ the HuggingFace training framework [80]. Eight A100 GPUs with 40 GiB of VRAM are used for training, while an H100 GPU is used for inference wall-time measurements. Experimental details of each subsection are summarized in Appendix G.

## 3.2 Main results

In Table 1, we measure the language modeling performance of the Block Transformer. Block models are scaled to have the same number of non-embedding parameters as the vanilla model variants. Our models, when having two or three times more parameters, achieve comparable perplexity and accuracy on five zero-shot evaluation tasks as the vanilla models. This is an expected result because two separate decoders spend fewer FLOPs per forward pass, reducing the attention complexity by a factor of $1/L_B^2$ at the block-level and by roughly $L_B/L$ at the token-level.

---

[5]During evaluation, we add left padding of length $L_B - 1$ to the first block. To use internal padding in blocks during inference, we apply random-length padding when packing documents for pretraining (see Appendix H). Absence of this technique results in significant performance drop for certain tasks such as LAMBADA.

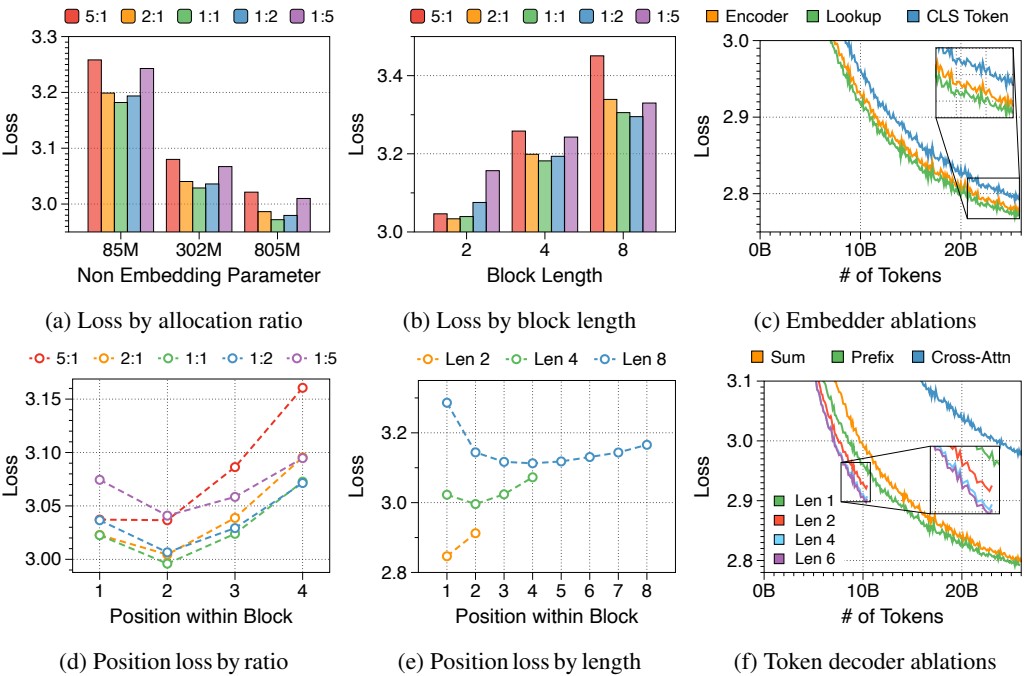

Figure 3: (Left: (a), (d)) Average and position-wise loss by the ratio of parameter allocation between block and token decoders. The ratio is represented as block to token decoders. (Center: (b), (e)) Average and position-wise loss in relation to block length $L_B$. (Right: (c), (f)) Training loss curve for variants of the embedder and token decoder. We consider four different lengths for the prefix-based token decoder. We use models with 302M non-embedding parameters and one-to-one ratio trained on 8 billion tokens.

The actual inference throughput and memory efficiency of the Block Transformer are significantly higher compared to vanilla models. We measure the maximum throughput [71], which use maximum batch sizes of each model variant allowed by memory. As shown in Figure 2a and Figure 2b, our models achieve Pareto-optimality, especially demonstrating up to 25 times increase, under two scenarios: *prefill-heavy* and *decode-heavy*, where the input and output sequence lengths are 2048, 128 and vice-versa. This efficiency improvement is due to effective reductions in KV cache memory, which allows batch sizes to be about six times larger, as summarized in memory per sample in Table 1. The Block Transformer further reduces latency in a prefill-heavy setting, as past KV states of prompts need to be cached only in the block decoder, without forwarding them to the token decoder. As detailed in Appendix I, this overall trend remains consistent even with the FlashAttention algorithm [23] employed during inference.

The Pareto frontiers for variable fixed batch sizes, i.e., 1, 32, and 256, are illustrated in Appendix J. We discover that as both the model size and batch size increase, the throughput rate of the Block Transformer scales exponentially. Considering that the LLMs typically utilized in real-world applications have billions of parameters, and taking into account the strategy of aggregating multiple user requests to optimize batch inference [42, 60, 71], the results suggest that our proposed architecture will demonstrate even more benefits in practical multi-tenant deployment scenarios.

In Figure 2c, we observe that the throughput of the Block Transformer with an 8K prompt length surpasses that of the vanilla model with a 2K prompt length. This is reasonable because the context length of the block decoder is reduced by a factor of 4, and the token decoder is nearly free of KV-cache overheads. Given the rising interest in enabling longer context lengths, even over one million tokens [15, 67, 52], the Block Transformer has potential to enhance throughput even further.

## 3.3 Analysis on parameter allocation ratio and block length

**Perplexity shows a U-shaped pattern across different allocation ratios** We explore the impact of different allocation ratios between the block and token decoders on language modeling performance,

while keeping the total number of non-embedding parameters constant. Figure 3a illustrates the training loss across five distinct ratios for three model sizes. Interestingly, there is a clear U-shaped trade-off at all three model sizes. We find that a one-to-one ratio is optimal for models with $L_B = 4$ consistently across all model sizes. If either side is too small, there is a noticeable decline in performance. This demonstrates the synergistic effect and the equal importance of the block and token decoders in language modeling.

**Larger block and token decoders reduce perplexity at initial and later positions respectively** We measure average loss at each position within a block, depicted in Figure 3d. The position-wise loss typically exhibits a U-shaped pattern, aligning with findings from a previous multiscale language model [84] and blockwise parallel decoding methods [72, 16, 41]. This trend stems from the lack of global context in context embeddings, which escalates uncertainty at later positions. Moreover, perplexity at specific positions correlates with the parameter sizes of two decoders. A larger block decoder significantly lowers initial position loss due to predictions solely based on the context embedding. In contrast, a larger token decoder improves prediction accuracy for later tokens by better leveraging local context. These interdependent effects dictate the optimal parameter ratio, with similar patterns evident in models of various sizes, detailed in Appendix K.

**Shorter block length favors larger block decoder whereas longer length prefers token decoder** Figure 3b demonstrates that training loss still follows a U-shaped pattern across different allocation ratios, regardless of block length. Optimal ratios shift with block length: shorter blocks benefit from a larger block decoder, while longer blocks perform better with more parameters in the token decoder. This is due to the inverse relationship between block length and FLOPs of the block decoder, which influences model capacity [25, 26, 34]. As Figure 3e shows, first position loss significantly decreases with shorter blocks, reflecting increased capacity in the block decoder. While the token decoder shows minimal differences in FLOPs across block lengths, it has more chance to improve the likelihood of later tokens as block length increases, favoring a larger token decoder. These trends are consistent across different model sizes and allocation ratios, detailed in Appendix L.

**Larger token decoder and longer block length are beneficial for achieving high-throughput** We evaluate the allocation ratio and block length from a throughput perspective, summarizing the Pareto frontier in Appendix M. Models with larger token decoders reach Pareto-optimality by achieving higher throughput at a minor performance compromise. Since KV cache IO significantly influences inference time, allocating more parameters to the token decoder is advantageous because the local context length is bounded by the block length. Additionally, increasing the block length improves throughput as KV cache length in the block decoder reduces proportionally. Therefore, although our main configuration uses a one-to-one ratio and a block length of four, opting for a longer block length and a larger token decoder could result in a higher-throughput model.

## 3.4 Ablation on components of the Block Transformer

**Lookup strategy is the most effective approach for the embedder** In Figure 3c, we experiment with three embedder strategies to bundle block tokens into a single embedding. Surprisingly, a complex transformer encoder like RoBERTa [47] does not outperform a simpler lookup table strategy. Moreover, the encoder-based embedder lowers generation throughput due to additional computational overhead. As a result, we opt for the lookup strategy to steamline the Block Transformer architecture. Although the CLS token approach allows flexibility in block length, we leave it for future work as it compromises language modeling performance.

**Prefix token decoder with longer prefixes enhances performance with minimal overhead** Figure 3f shows the training loss curve for three token decoder strategies. Using a cross-attention module with key and value sequences equal to the block length considerably diminishes performance. In contrast, forwarding context embeddings through self-attention operations enhances performance, with prefix decoding surpassing other methods. Furthermore, extending the prefix beyond four tokens markedly improves perplexity, effectively broadening the computation width of token decoder. Since longer prefixes add minimal inference overhead, we select a prefix length of two by balancing performance with FLOPs. This approach offers new insights into global-to-local modeling, diverging from previous studies [84] which overlook the potential of local computational capacity in the token decoder. Detailed results across various model sizes are summarized in Appendix N.

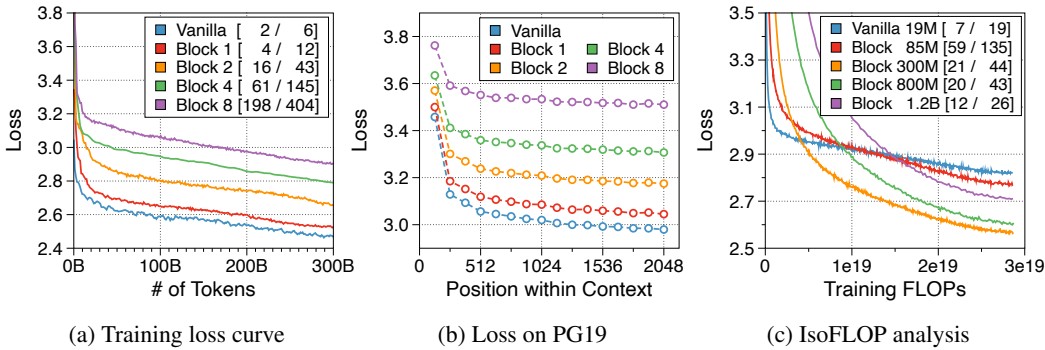

(a) Training loss curve      (b) Loss on PG19      (c) IsoFLOP analysis

Figure 4: (a) Training loss curve with varying block lengths. The numbers in the brackets represent the maximum throughput, measured in 1K tokens per second, for prefill-heavy and decode-heavy settings, respectively. (b) The loss at different token positions within context length on the PG19 test set. We average over every 128 sequences for smoothing. (c) Training loss curves under the same budget for both training FLOPs and inference throughput.

## 3.5 Analysis on global-to-local language modeling

**Global-to-local language modeling efficiently optimizes throughput relative to performance** In Figure 4a, we transition from vanilla to Block Transformers by adjusting block lengths. As block length increases, training loss changes log-linearly and throughput increases exponentially, clearly demonstrating the efficiency of global-to-local modeling. Using a lookup embedder and token decoder with one prefix token, our model with $L_B = 1$ differs from the vanilla model only by removing global attention in the upper layers. Notably, this model achieves loss equivalent to that of the vanilla model after training on 70% of the tokens, while doubling throughput. Despite pruning all past sequences, this robust performance shows that the context embedding can retain relevant information, enabling the effective of use local computations in global-to-local language modeling.

**Block transformer can effectively leverage full context** Since the token decoder depends solely on the context embedding, there could be a concern about whether the Block Transformer fully utilize context information. To address this, we evaluate the loss of token positions within a 2K context window using the test set of PG19 dataset [62]. Figure 4b indicates that later tokens are consistently predicted with higher likelihood, suggesting that our architecture, which distinguishes between block-level and token-level decoders, effectively leverages full context information. Furthermore, we observed the same trend with an 8K context length (in Figure 20), demonstrating our model's ability to fully exploit at least 8K tokens of context. Our block language modeling further demonstrated its effective utilization of full context in the recent Needle-In-a-Haystack long-context benchmark [39] (refer to Appendix O for details.)

## 3.6 IsoFLOP analysis under inference throughput constraints

Previous studies have focused on compute-optimal models to maximize performance within training FLOPs budgets [40, 37], while typically overlooking inference throughput. Recent trends, however, emphasize models that also consider inference throughput constraints, either by overtraining smaller models [76, 74] or by reducing FLOPs of the model itself [65]. In Figure 4c, an optimal Block Transformer model achieves superior perplexity and triples the throughput when using the training FLOPs and throughput of the vanilla model as budget constraints. This illustrates that our models can effectively balance training efficiency and inference throughput.

## 3.7 Uptraining from vanilla transformers

Unlike previous studies [84], our subword-level global-to-local architecture can leverage the initialization from a pretrained vanilla transformer. This enables efficient training, requiring only a small number of data. As shown in Figure 5a, this uptraining strategy can lead to near-full performance recovery with just 10% of the original training steps, outperforming random initialization strategy. Consistent with previous studies [2, 5], investigating deliberate weight initialization techniques can further enhance the performance convergence. We summarize details in Appendix P.

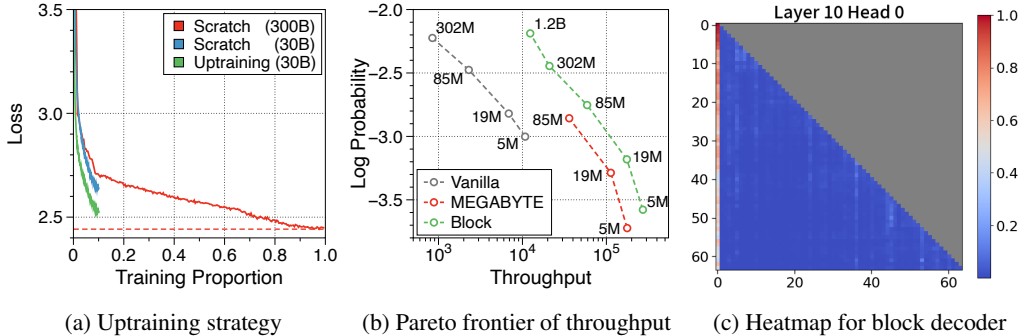

| (a) Uptraining strategy | (b) Pareto frontier of throughput | (c) Heatmap for block decoder |

Figure 5: (a) Training loss curve with uptraining strategy. The red horizontal line refers to the training loss of a full pretrained model. (b) Throughput comparison to MEGABYTE. We compare to three sizes of MEGABYTE in the prefill-heavy setting. (c) Visualization of heatmap for attention scores in block decoder. We visualize only the first 64 sequences for clarity.

## 3.8 Comparison to related works

**Performance comparison to MEGABYTE**    The MEGABYTE model [84] adopts a global-to-local structure but focuses on efficient pretraining over inference. Thus, within the training FLOPs budget, they argue for a larger block decoder based on a 6:1 ratio deemed optimal. As shown in Figure 5b, we reimplement the token-level MEGABYTE models, and they also achieve significantly higher throughput compared to vanilla models through global-to-local modeling. Nevertheless, consistent with our insights in Section 3.3, our models with enhanced local computational capacity demonstrate a significant throughput increase of over 1.5 times on top of MEGABYTE. See Appendix Q for more details.

**Relation to KV cache compression**    Global-to-local modeling can be viewed through the lens of KV cache compression, where past sequences are entirely pruned (yet compressed into a context embedding) in the upper layers. Similar to techniques that preserve only meaningful tokens determined by accumulated attention scores [78, 87], this offers a promising way to improve decoding speed without compromising performance. Following the observations of [82, 32] that attention often concentrates on the first token (frequently semantically unimportant), Figure 5c reveals a similar pattern in our block decoder. This suggests that augmenting the token decoder's input with the initial "sink" block embedding, or a local window of embeddings, could yield substantial performance gains.

## 4 Discussion

### 4.1 Contextual information encapsulated in context block embedding

Since the input tokens and context embeddings share the same latent space in the token decoder, we analyze the nearest tokens to these block embeddings. Interestingly, Table 7 in Appendix S reveals that context embeddings compress global context rather than outlining the next block. The second prefix often contains information about the last token of current block to aid predicting the first token of the next block. Meanwhile, the first prefix typically matches non-intuitive or the EOS token, suggesting that they carry more general information. In light of this, the block decoder effectively compresses past global contexts, which the token decoder leverages for its local language modeling.

### 4.2 Techniques for further throughput improvement

**Block autoregressive model with parallel token decoding**    When we pretrain the block decoder to predict next input block embeddings, the token decoder can decode all blocks in parallel if the predictions from block decoder are precise. While Mujika [50] enhance pretraining efficiency by directly predicting the embedding matrix, we find that MSE or contrastive losses [18] at the block decoder actually degrades performance. Moreover, error accumulation at the block level needs to be addressed, as discretization is not possible with block embeddings. Nevertheless, using pretrained text embeddings [79, 43] as ground truth, instead of jointly training embedder, could be beneficial.

**Predicting multiple blocks at once with longer output length**   If the model is trained to predict two or three blocks simultaneously, throughput will increase proportionally. For example, if the input block length is four, the token decoder can be pretrained to predict eight tokens, equivalent to two blocks. One efficient training method could be uptraining the original Block Transformer models. To guarantee performance, we can adaptively adjust the prediction length based on the confidence of subsequent blocks or verify those drafts, similar to speculative decoding [44, 17, 46].

## 5   Related work

The Block Transformer exemplifies an approach to *coarse* and *local* processing in autoregressive transformers. We compare our architecture with previous approaches for coarse and local processing, underscoring its unique effectiveness in alleviating bottlenecks in autoregressive inference.

**Hierarchical transformers for coarse processing**   Many previous works have explored hierarchical transformers to process long sequences efficiently. Early works use local encoders at early layers to obtain pooled representations of documents [59] or image patches [36]. Later works explore a downsample-then-upsample approach to process long sequences at coarser levels within the core of the model [22, 53]. This has been followed by numerous improvements including an autoregressive formulation [56], dynamic pooling lengths [57], and reduced decoding steps [28]. While this can reduce KV cache at middle layers, the upper and lower layers suffer from the same attention bottlenecks as vanilla transformer layers, where representations are at the finest level. In contrast, we identify these bottlenecks and mitigate them by applying local processing at the finer levels.

**Local processing in modern language models**   Early hierarchical transformers employ local computation at early layers [59, 36], but these are limited to encoder models. The most prominent adaptation of local processing in modern LMs is sliding window attention (SWA) [19, 8], adopted in GPT-3 [14] and Mistral [38]. While SWA reduces KV cache memory, window sizes are typically much longer than the block lengths used in Block Transformer; Mistral uses a window size of 2048, which requires 512 times more KV cache compared to our baseline token decoder. Previous adaptations of SWA typically incorporate global attention layers [14] or exploit stacked layers [38] to attend to older sequences. For example, with window size $W$, the current token attends to $W$ tokens of context in the final layer; these attend to up to $2W - 1$ tokens in the previous layer; and so on. Due to this dependency, SWA does not benefit from the same prefill optimizations as the Block Transformer. Under our global-to-local approach, the token decoder restricts attention within block boundaries *across all layers*, enabling the prefill stage to be skipped for all preceding blocks.

**Global-to-local hierarchical transformers**   Several works on byte-level modeling employ a similar structure as our Block Transformer [84, 50]. However, while we aim to optimize autoregressive inference for subword-level LMs, prior work mainly utilize the hierarchical structure to mitigate the long context lengths of byte-level data in the absence of tokenization. Hence, in contrast to the central role of our local module (token decoder), prior work consider the role of their local module as 'mapping a hidden state to a distribution over possible patches', and suggest that 'much smaller model can be used' [84] and may 'cease to contribute to overall performance' [50]. Yu et al. [84] concludes that it is optimal to assign more parameters to the global module under *fixed training-time* constraint. In contrast, we find that a more balanced allocation, e.g., 1:1, performs better *and* faster under *fixed parameter* constraints, and that even larger token decoders can maximize *inference throughput*, showing that the local module can contribute to performance in an efficient manner. By recognizing the key bottlenecks in autoregressive inference, we believe our work presents a novel interpretation and uncovers previously unrecognized benefits of global-to-local hierarchical transformers.

## 6   Conclusion

We introduced the Block Transformer architecture which highlights the inference-time advantages of global-to-local modeling in autoregressive transformers. Our empirical findings demonstrate that both global and local components play vital roles, and we recognize the inference benefits of token decoder, which was overlooked in previous work. By strategically designing our architecture, we significantly improve throughput compared to vanilla transformers of equal performance. Refer to Appendix A for limitation, Appendix B for future works, and Appendix C for broader impacts.

## Acknowledgments

We would like to thank Honglak Lee for his critical feedback on the outline of the paper. We extend our gratitude to Yujin Kim for extensive discussions on efficient inference and related work. Additionally, we appreciate the continuous feedback from Kyungmin Lee, Junwon Hwang, Sangha Park, and Hyojin Jeon, throughout the development of our work. Thanks to Joshua Ainslie for discussion during the early stages of development.

This work was supported by Institute of Information & communications Technology Planning & Evaluation (IITP) grant funded by the Korea government (MSIT) (No.2019-0-00075, Artificial Intelligence Graduate School Program (KAIST), 10%) and the Institute of Information & communications Technology Planning & Evaluation (IITP) grant funded by the Korea government (MSIT) (No. 2022-0-00871, Development of AI Autonomy and Knowledge Enhancement for AI Agent Collaboration, 90%).

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

# Contents

# A   Limitations

The Block Transformer variants considered in our study require more parameters and FLOPs compared to their perplexity-equivalent vanilla models. Despite higher parameter and FLOP requirements, our Block Transformers achieve higher inference throughput, owing to low memory overhead and omission of prefill in the token decoder. However, this advantage is diminished during training–resulting in higher wall-time training costs compared to vanilla Transformers. The large parameter requirements also hinder the applicability of Block Transformers in situations with hard memory constraints such as on-device usage. We note that these are partially a result of our focus on inference throughput, rather than architectural limitations. There are many promising avenues to minimize parameter and FLOP (training cost) requirements, with minor adjustments to the architecture or hyperparameters. In the following section, we discuss several of these for future work.

# B   Discussion and future works

## B.1   Optimizing hyperparameters for parameters or FLOPs

We can optimize the hyperparameters of the Block Transformer architecture to minimize parameter or FLOP requirements, as opposed to inference throughput as in our main experiments. Frist, we can reduce the *block length* to enhance performance while maintaining the same parameter count. Our ablations on block length demonstrate that a shorter block length can significantly improve perplexity, while compromising inference throughput with increased FLOPs in the block decoder. Thus, to achieve comparable perplexity, we can utilize less parameters, which offsets the decreased throughput resulting from the shortened block length.

Secondly, we find that increasing the proportion of the block decoder can significantly reduce FLOP requirements with minor degradation in performance, due to the FLOP-intensive nature of the token decoder. However, this comes at the cost of increased inference wall-time due to the KV cache bottlenecks of the block decoder. Further experimentation is needed to precisely identify the tradeoffs associated with these hyperparameter choices with respect to various cost metrics.

## B.2   Densification of the block decoder with longer block embedding

Another approach to improving the performance of Block Transformers without extra parameters would be through better utilization of those already in the block decoder, i.e., by passing more tokens through them. We could do this by representing a single block with a longer input block embedding, say $L_B$, instead of one. Let's call these *subblock tokens*. During a single decoding step, $L_B$ input tokens would be projected into $L_B$ subblock tokens. Then, these would be passed to the block decoder and forwarded in parallel.

This would effectively preserve the computational width [34] of the block decoder, i.e., the total embedding dimension of the inputs, to be equivalent to a vanilla Transformer of the same width and depth. The minor difference in perplexity between the vanilla Transformer and Block Transformer with $L_B = 1$ in Figure 4a suggests that Block Transformers could approach the performance of same-sized vanilla transformers when the computational width of the block decoder is the same.

While this would require the same FLOPs as a vanilla Transformer, we can expect roughly $L_B$ times reduction in decoding wall-time due to parallel execution—since parameters and previous KV cache would only need to be fetched once per block, instead of once per input token. Note that total KV cache storage would be the same as vanilla Transformers since the number of input tokens and subblock tokens would be the same (this is why we expect $L_B$ reduction in KV cache IO rather than $L_B^2$ as in our original block decoder).

## B.3   Relieving the locality of the token decoder for performance gains

In our experiments, we bottleneck the global information passed to the token decoder into a single context embedding. This is done for simplicity and to highlight the viability of global-to-local modeling, where the local module has limited access to global context. However, we posit that the token decoder can benefit from performance gains with minimal extra costs by relieving this rather extreme limitation.

It is possible use additional context embeddings in the token decoder to facilitate the propagation of context information, as discussed in Section 3.8. Instead of projecting only the last output block embedding to the token decoder, we could utilize a small window of previous output block embeddings. This could resolve the rise in perplexity in later positions in the token decoder due to insufficient context information, with only slight increase in FLOPs and KV cache overhead in the token decoder.

### B.4  Further scaling and advanced uptraining schemes

The scale of experiments in our paper is relatively small compared to even previous-generation frontier models [14, 20]. While our experiments show that the inference throughput benefits of Block Transformers scale positively across two orders of magnitude, further experiments are required to verify this beyond 1 billion parameters.

We can consider uptraining as a cost-effective training approach for this analysis, which effectively utilizes existing pretrained vanilla transformers to minimize the training costs of Block Transformers. For example, we can consider a progressive adaptation approach where a vanilla transformer is first adapted to a Block Transformer with block length 1, to maximize compatability, and then progressively trained with larger block lengths. Moreover, instead of simply splitting the layers of a pretrained vanilla transformer to initialize the block and token decoders, exploring weight initialization methods like averaging the layers or identifying weights that produces similar activations could significantly enhance performance.

### B.5  Adaptive block lengths for dynamic compute allocation

What if we can dynamically allocate computation to generate 'easy' tokens faster but ponder longer on 'hard' tokens? This has been the central question of several previous works on dynamic compute allocation [35, 68, 4, 65]. The multiscale nature of the Block Transformer architecture offers a novel avenue to achieving this in autoregressive language models–by dynamically setting the input and output length of blocks based on the 'difficulty' of its contents. For the embedder and token decoders, we can use our CLS-token and prefix token based designs respectively, and padding can be used to maintain static computation during training. A challenge remains in training the model to dynamically determine optimal input *and* output block lengths.

## C  Broader impact

Recent language models have been scaled up significantly to achieve human-like capabilities, resulting in substantial training costs. Deploying these extensively large models in real-world services incurs significant computational overhead. Moreover, the escalating computational costs associated with large language models are raising environmental concerns. Our model enhances memory utilization and inference throughput, potentially mitigating these issues. The efficiency gains from the Block Transformer architecture can reduce the cost of deploying language models. Additionally, the global-to-local modeling at the subword level facilitates efficient uptraining from existing pretrained models to Block Transformers, providing a training-efficient pathway for enhancement. We encourage further research to fully explore these impacts, ensuring responsible development and deployment of Block Transformers.

## D  Extended related work

### D.1  KV cache compression

Recent advancements in KV cache compression aim to optimize memory usage by selectively retaining essential key-value pairs [82, 87, 32, 83, 45]. Scissorhands [32] and H2O [87] enhances compression by leveraging attention scores to preserve only the crucial components of the KV cache. Fast-Gen [48] refines this approach by employing distinct policies per attention head. StreamingLLM [82] maintains only the recent context window and a few initial tokens as an 'attention sink', thereby discarding other past context. SnapKV [45] focuses on pruning tokens in the input prompt, in response to increasing input lengths. PyramidInfer [83] prunes KV heads during prefill, as each layer is computed,

to tackle memory usage in this stage. While various methods have been proposed to intelligently prune tokens that are relatively less important, these approaches essentially permanently discard information which may become relevant again in future contexts. In contrast, Block Transformer retains access to all previous context in the block decoder. KV cache compression methods can also be applied to the block decoder to improve efficiency.

### D.2 Architectural for optimizations of KV cache

Recent works modify the design of the attention block such that multiple query heads can attend to the same shared KV heads, significantly reducing the number of unique KV heads while minimal degradation in performance. Multi-query attention (MQA) [70] allows multiple query heads to attend to shared key/value pairs, reducing storage overhead. Grouped-query attention (GQA) [2] generalizes this by organizing query heads into groups sharing a single KV head to achieve the same goal. Several concurrent works take this idea even further, by sharing KV heads between adjacent layers [13] or share the KV head of the top layer across the majority of layers [81]. A recent architecture [24] introduces multi-head latent attention (MLA) to jointly quantize KV states. By adopting standard transformer architectures, our Block Transformer can also benefit from these techniques to mitigate the remaining KV cache bottlenecks in the block decoder.

Several works take novel approaches to the overall architectural formulation. Tandem Transformers [55] alternate between a *large* block-level encoder and *small* token-level decoder. YOCO [73] is a decoder-decoder architecture that employs a cross-attention based decoder at upper layers which all refer to KV cache from a single middle layer which mitigates KV cache storage. In contrast, we take a different approach where the context information is compressed into a single context embedding to enable local modeling, nearly free of KV cache storage *and* access costs, mitigating critical bottlenecks in inference throughput.

## E  Analysis on the inference efficiency of Block Transformer

### E.1  Background: inference stages and principal bottlenecks

To generate a response to an input prompt, it is necessary to prefill and cache the KV values of all input tokens, as they are attended by subsequent tokens under global self-attention. (1) The prefill phase is computation-bound because all input tokens can be processed in parallel during one forward pass. In contrast, when generating new tokens, only a single token can be processed per forward pass, as the output of the previous token is needed as the input for the next. While linear projection FLOPs are dominant with short context lengths, self-attention FLOPs surpass linear projection FLOPs with very large context lengths, due to quadratic scaling. (2) The decode phase is memory access-bound because all model parameters and previous KV cache must be loaded from memory at *each forward pass*. To achieve high compute utilization and throughput, production serving systems typically leverage batching to amortize the cost of parameter IO [1, 51]. Thus, under large batch sizes (and sufficiently long contexts), KV cache IO becomes the main bottleneck in decoding [61].

### E.2  Inference-time advantages of block and token decoders

The following paragraphs provide a detailed presentation of the inference benefits associated with our proposed block and token decoder. For visual clarification, refer to Figure 6, and Table 2 offers a comparative analysis of actual computation speeds.

**Block decoder reduces prefill computation by $L_B$ and decode IO by $L_B^2$**  The block decoder maintains global attention similar to vanilla transformers but operates at a much coarser block level, reducing context length by $L_B$ compared to the original token-level sequence. This reduction decreases position-wise computation during prefill by $L_B$ compared to vanilla transformers of the same size. The main bottleneck during batch decoding, i.e., KV cache IO, is reduced by $L_B^2$ as it is quadratic to context length. The same savings apply to attention computation, which can become a bottleneck during prefill as context lengths grow. KV cache storage in GPU memory during decoding is also reduced linearly by $L_B$, enabling larger batch sizes and higher parallelism.

**Token decoder skips prefill entirely and nearly eliminates decode IO**   The token decoder does not use global attention but relies on a single context embedding for global context information, applying attention within each independent block for local context. Thus, the token decoder does not need to preserve or retrieve KV cache values from previous blocks, *eliminating the need to prefill input tokens*. This also nearly eliminates KV cache IO overhead during decoding, as quadratic scaling applies to the small local context of $L_B$ rather than the global context $L$. Compared to the KV cache IO complexity of $L^2$ in vanilla transformers, token decoders have $L_B^2$ complexity per block, across $L/L_B$ blocks, achieving an overall reduction of $L/L_B$. For our main models with $L = 2048$ and $L_B = 4$, this results in a *256-fold reduction in KV cache IO overhead*. Asymptotically, this reduces KV cache IO overhead from quadratic to linear with respect to context length, solving a key challenge in scaling to very long contexts [29]. KV cache storage is also reduced by the same factor, enabling larger batch sizes. This significantly improves the utilization of inference hardware, which is typically as low as ∼1% model FLOPs utilization (MFU) in vanilla transformers [61]. Thus, we can apply more FLOPs in the token decoder to improve performance, with minimal effect on inference throughput.

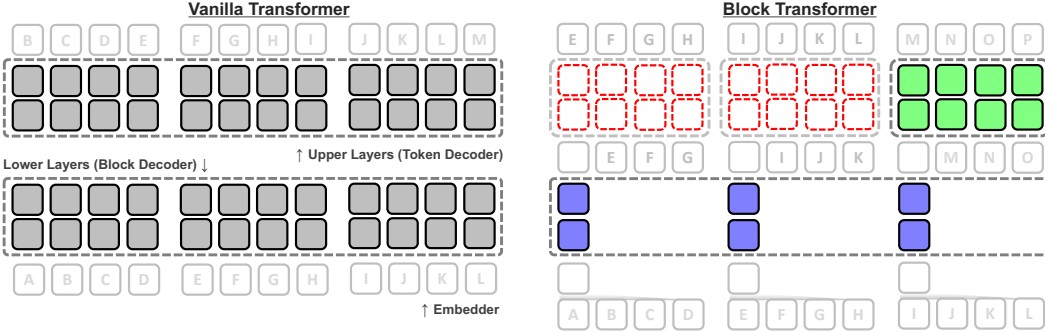

Figure 6: Illustration of key advantages of Block Transformer over Vanilla Transformers. Each colored box represents a single input unit that is processed at each layer, and input tokens A to L are prompt tokens.(1) The local token decoder does not need to prefill the prompt, as it is not used in subsequent generation, whereas the upper vanilla layers require the KV values of *all* prompt tokens to generate token M and onwards. (2) The token decoder only needs to fetch KV cache from up to $L_B = 4$ local tokens, while the upper vanilla layers needs to fetch from all previous tokens *at each step*, which can go up to the thousands or millions. (3) Since the block decoder operates at the block level, overall computation, memory overhead, and forward steps are reduced by $L_B = 4$. KV cache IO is reduced quadratically. (4) Block transformer enables significantly higher batch size and thus overall higher compute utilization on identical hardware, as it only needs to preserve the KV cache of the blue and green parts in memory (green is minuscule for longer context lengths).

Table 2: Measurements on key advantages of Block Transformer during prefill and decode. Per-sample walltime of key operations at lower layers (block decoder) and upper layers (token decoder), for vanilla model with 300M non-embed params and a better performing Block Transformer with 1.2B non-embed params, using identical hardware (one H100 GPU). Refer to the caption of Figure 1 on the reason behind the significant walltime savings, despite using more parameters.

|  |  | Vanilla T. | | Block T. | |
|---|---|---|---|---|---|
|  |  | Prefill | Decode | Prefill | Decode |
| Upper Layers | Attention | 19.94 | 56.21 | N/A | 2.41 |
| (Token Decoder) | FFN | 0.86 | 1.42 | N/A | 0.55 |
| Lower Layers | Attention | 19.94 | 56.21 | 1.96 | 5.00 |
| (Block Decoder) | FFN | 0.86 | 1.42 | 0.68 | 0.09 |

(a) Prefill-heavy setting (2048/128)

|  |  | Vanilla T. | | Block T. | |
|---|---|---|---|---|---|
|  |  | Prefill | Decode | Prefill | Decode |
| Upper Layers | Attention | 0.15 | 500.50 | N/A | 31.55 |
| (Token Decoder) | FFN | 0.06 | 16.75 | N/A | 7.71 |
| Lower Layers | Attention | 0.15 | 500.50 | 0.05 | 47.24 |
| (Block Decoder) | FFN | 0.06 | 16.75 | 0.04 | 1.44 |

(b) Decode-heavy setting (128/2048)

# F   Architectural details

## F.1   Embedder methods

**Lookup**   For our main embedder design, we simply retrieve token-level embeddings from a lookup table and concatenate them to obtain the input block embeddings. The token-level embedding dimension is set to be $1/L_B$ of the main model dimension.

**Encoder**   To ablate the effect of adding encoding capability to the embedder, we encode the input tokens of a block with a small RoBERTa-based encoder. We use a fixed sized encoder with dimension size of 256 and 3 hidden layers. We concatenate the output hidden states and apply linear projection to obtain the input block embedding.

**CLS token**   To investigate the feasibility of an embedder that can accept various input block lengths, we use CLS tokens previously used to extract sentence embeddings [27]. We use the same model size as the RoBERTa model and encode information in 3 CLS tokens, to increase the embedding dimension while minimizing the model dimension of the embedder. Similar to the RoBERTa embedder, we concatenate the output hidden states of the CLS tokens and apply linear projection to obtain the input block embedding.

## F.2   Token decoder methods

**Prefix**   For the main token decoder design, we incorporate the context embeddings from the block decoder by projecting them as prefix token embeddings. The token decoder can retrieve the context information from the prefix tokens via attention, and also further encode the context information. We can use multiple prefix tokens, i.e., increase the prefix length, to increase the computational width [34] of the token decoder to increase performance with addtional FLOPs, are relatively cheap in terms of inference time in the token decoder.

**Summation**   We also consider the summation method used in previous work [84]. Here, the context embeddings are projected to $L_B$ embeddings of dimension $D$ and added to the token embeddings at each input position of the token decoder. This does not benefit from additional computation of the context information in the token decoder.

**Cross-attention**   Finally, we consider an approach that uses cross-attention, treating the output context embedding as the output hidden states of an encoder in an encoder-decoder transformer [63]. Specifically, we project the the context embedding into $L_B$ hidden states each with dimension $D$ and apply cross-attention between self-attention and feedforward operations at each transformer layer in the token decoder. This also does not benefit from additional computation of the context information in the token decoder.

# G   Experimental settings

## G.1   Overall settings

We use the same transformer architecture as Pythia [10], utilizing the open-source GPT-NeoX library [3]. We train both vanilla and Block Transformer models on the Pile [30, 9], which is a curated collection of English datasets specifically developed for training large language models. We utilize a BPE tokenizer tailored for the Pile dataset [12], including a vocabulary size of 50,304. The models are pretrained on approximately 300 billion tokens, which corresponds to about 1.5 epochs of training, given that the deduplicated Pile comprises 207 billion tokens. To evaluate the models on various zero-shot tasks, we use the Language Model Evaluation Harness framework [31]. We employ the HuggingFace training framework [80] and enhance memory efficiency through mixed precision training and the Zero Redundancy Optimizer (ZeRO) [64] from the DeepSpeed library [66]. We use eight A100s with 40 GiB of VRAM for training, while we measure the inference latency using an H100 GPU.

## G.2 Model sizes and hyperparameters

Our models are trained across six different sizes, varying from 33 million (M) to 1.4 billion (B) parameters, to explore how performance scales with model size. We train four vanilla models corresponding to our Block Transformer models. We summarize detailed model configurations and training hyperparameters in Table 3.

Table 3: Hyperparameters for vanilla and block models. The size of each model refers to the size of non-embedding parameters. The transformer in vanilla model are summarized under the token decoder. $n_L$ denotes the number of layers, and $L$ and $L_B$ represents the context length and block length, respectively. For the token decoder, $L_{ctx}$ is calculated by summing the prefix length of two and the block length of four. We note that the lookup method is used as the embedder component.

| Models | Size | Token Decoder | | | | | Block Decoder | | | | | LR | Batch |
|---|---|---|---|---|---|---|---|---|---|---|---|---|---|
| | | Method | $L$ | $n_L$ | Dim | Head | $L_B$ | $L$ | $n_L$ | Dim | Head | | |
| Vanilla | 5M | - | 2048 | 6 | 256 | 8 | - | - | - | - | - | 1e-3 | 256 |
| | 19M | - | 2048 | 6 | 512 | 8 | - | - | - | - | - | 1e-3 | 256 |
| | 85M | - | 2048 | 12 | 768 | 12 | - | - | - | - | - | 6e-4 | 256 |
| | 302M | - | 2048 | 24 | 1024 | 16 | - | - | - | - | - | 3e-4 | 256 |
| Block | 5M | Prefix | 2+4 | 3 | 256 | 8 | 4 | 512 | 3 | 256 | 8 | 1e-3 | 256 |
| | 19M | Prefix | 2+4 | 3 | 512 | 8 | 4 | 512 | 3 | 512 | 8 | 1e-3 | 256 |
| | 85M | Prefix | 2+4 | 6 | 768 | 12 | 4 | 512 | 6 | 768 | 12 | 6e-4 | 256 |
| | 302M | Prefix | 2+4 | 12 | 1024 | 16 | 4 | 512 | 12 | 1024 | 16 | 3e-4 | 256 |
| | 805M | Prefix | 2+4 | 8 | 2048 | 16 | 4 | 512 | 8 | 2048 | 16 | 3e-4 | 512 |
| | 1.2B | Prefix | 2+4 | 12 | 2048 | 16 | 4 | 512 | 12 | 2048 | 16 | 2e-4 | 512 |

## G.3 Settings for Section 3.2

Each model is trained for 300 billion tokens with a context length of 2048. For the Block Transformer models, we set the block length to four, and leverage prefix decoding with a length of two and lookup methods as the token decoder and embedder components, respectively. To measure the allocated memory and throughput, we use synthetic samples where all prompts are padded to the target length.

## G.4 Settings for Section 3.3

Unless otherwise specified, we use a default setting of a model with 302M non-embedding parameters, allocating the same size of parameters to both the block and token decoders. For the default strategies of embedder and token decoder components, we use three CLS tokens from a RoBERTa model, composed of three layers with a dimension of 256, and a prefix with a length of one, respectively. Extensive experiments reveal that finding the optimum requires minimal overhead because the ranking trend between ablations remains consistent from the early training stages, across various model sizes. Therefore, we train the models with just 8 billion tokens.

## G.5 Settings for Section 3.4

Each model is trained with a block length of four on 26 billion tokens, with the parameters of the block and token decoder being distributed equally. We have experimented with two model sizes of 85M and 302M non-embedding parameters. We set the default strategy for the embedder as utilizing three CLS tokens from the RoBERTa model, composed of three layers with a dimension of 256, and for the token decoder as prefix decoding with a length of one.

## G.6 Settings for Section 3.5

We use both vanilla and Block Transformers with the non-embedding parameters of 85M. All models are fully pretrained on 300 billion tokens with a context length of 2K. For Block Transformer models, we use a lookup strategy and prefix decoding with a length of one to facilitate a smooth transition from vanilla models to Block Transformers.

### G.7  Settings for Section 3.6

We train Block Transformer variants using the training FLOPs and inference throughput of a vanilla 70M model as constraints. All models are pretrained from scratch, with their training steps adjusted to match their respective FLOPs. The learning rate has fully decayed at the end of training steps.

### G.8  Settings for Section 3.7

To leverage the pretrained layer weights of the vanilla transformer model, we allocate parameters equally to the block and token decoders, preserving the overall non-embedding parameter size. Additionally, after concatenating four token embeddings from a lookup table of the vanilla models, we introduce a fully-connected layer to map it into the hidden dimension of the block decoder. We evaluate two models with 85 million and 302 million non-embedding parameters, training them on 30 billion tokens (10% of the original training data).

### G.9  Settings for Section 3.8

**Performance comparison to MEGABYTE**   We have reimplemented several variations of the MEGABYTE model, with their configurations detailed in Table 4. MEGABYTE bases its model dimensions on the GPT-3 model configuration [14] and argues that a block and token decoder parameter ratio of approximately 6:1 is optimal when considering training FLOPs budgets. We pretrained these models from scratch on 300 billion tokens.

Table 4: Hyperparameters for various sizes of MEGABYTE models. The size of each model refers to the size of non-embedding parameters. $n_L$ denotes the number of layers, and $L$ and $L_B$ represents the context length and block length, respectively.

| Models | Size | Token Decoder | | | | | Block Decoder | | | | | LR | Batch |
|---|---|---|---|---|---|---|---|---|---|---|---|---|---|
| | | Method | $L$ | $n_L$ | Dim | Head | $L_B$ | $L$ | $n_L$ | Dim | Head | | |
| | 5M | Sum | 4 | 4 | 128 | 4 | 4 | 512 | 5 | 256 | 8 | 1e-3 | 256 |
| MEGABTYE | 19M | Sum | 4 | 4 | 256 | 8 | 4 | 512 | 5 | 512 | 8 | 1e-3 | 256 |
| | 85M | Sum | 4 | 4 | 512 | 8 | 4 | 512 | 11 | 768 | 12 | 6e-4 | 256 |

**Relation to KV cache compression**   To explore attention scores, we utilize a pretrained Block Transformer model with 1.2B non-embedding parameters. The attention scores are extracted from randomly selected samples. Furthermore, we focus on the first attention head of each of the 12 layers in both the block and token decoders.

## H   Random length padding during pre-training

To apply inference on prompts whose lengths are not multiples of $L_B$, we need to add padding tokens to the prompt to fill the input blocks. Unlike padding tokens in vanilla transformers, these padding tokens are actually considered in the computation of the input block embedding, due to the fixed-size nature of our embedding methods, except for the CLS token variant. Therefore, we add random padding tokens with uniform length between 0 and $L_B - 1$ at the beginning of each document when applying input packing during pre-training. We also pad the unfilled tokens in the last block of each document, to prevent multiple documents being included in a single block. Note that this was applied after our main experiments, thus were not applied to our largest models in Table 1. We posit that this has adversely affected some downstream task performance evaluations. Figure 7 presents a comprehensive overview of the results obtained with and without appending random padding during both training and inference stages.

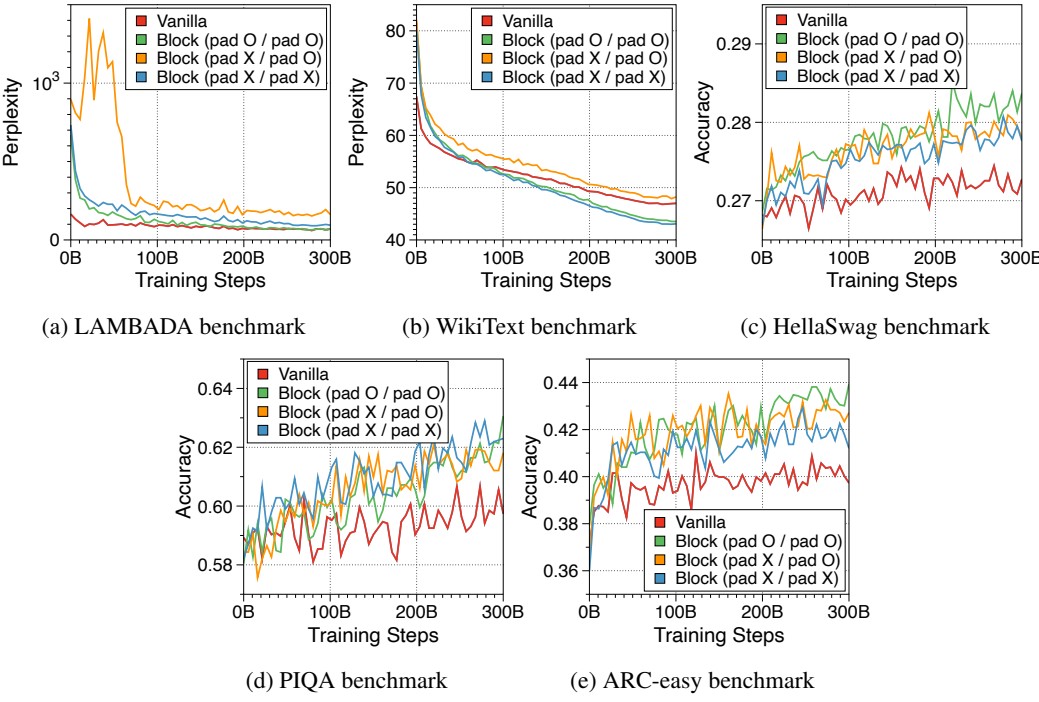

Figure 7: Zero-shot evaluation performance of vanilla and Block Transformer models. We use a 19M vanilla model and a 85M Block Transformer model. The first 'pad' in parentheses indicates whether random-length padding is used for input packing during training, and the second 'pad' indicates whether $L_B - 1$ length of padding tokens are added before the first token during inference.

# I Throughput Comparison with FlashDecoding

In modern LLM deployments, decoding speed enhancements through kernel fusion techniques like the FlashAttention algorithm [23] are generally employed. These mechanisms reduce the number of memory accesses, leading to faster decoding in LLMs. Consequently, the speed advantages of our Block Transformer, which minimizes KV cache size and memory accesses, could be a little diminished compared to a vanilla Transformer. To investigate this, we measured the maximum throughput with FlashDecoding applied, as illustrated in Figure 8. Interestingly, we observed an overall trend similar to that presented in Figure 2 in the main paper. Our model architecture still benefits significantly from FlashAttention for global attention within the block decoder, resulting in a considerable speed improvement of up to 31%.

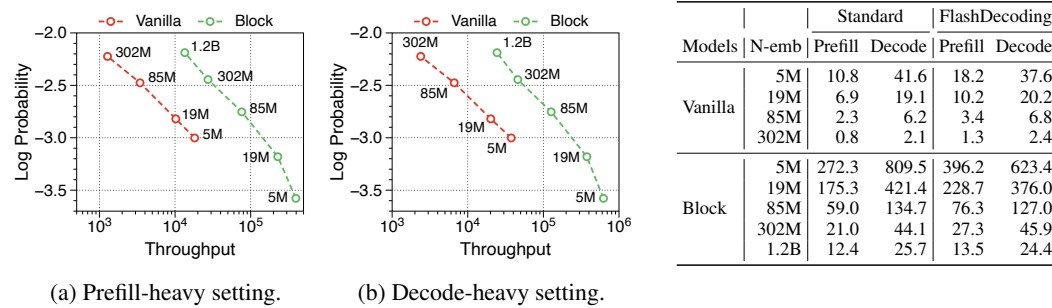

| Models | N-emb | Standard | | FlashDecoding | |
|---|---|---|---|---|---|
| | | Prefill | Decode | Prefill | Decode |
| Vanilla | 5M | 10.8 | 41.6 | 18.2 | 37.6 |
| | 19M | 6.9 | 19.1 | 10.2 | 20.2 |
| | 85M | 2.3 | 6.2 | 3.4 | 6.8 |
| | 302M | 0.8 | 2.1 | 1.3 | 2.4 |
| Block | 5M | 272.3 | 809.5 | 396.2 | 623.4 |
| | 19M | 175.3 | 421.4 | 228.7 | 376.0 |
| | 85M | 59.0 | 134.7 | 76.3 | 127.0 |
| | 302M | 21.0 | 44.1 | 27.3 | 45.9 |
| | 1.2B | 12.4 | 25.7 | 13.5 | 24.4 |

(a) Prefill-heavy setting.     (b) Decode-heavy setting.

Figure 8: Pareto frontier of throughput to langauge modeling performance using FlashDecoding.

# J Pareto frontiers at variable batch sizes and context lengths

In Figure 9 and Figure 10, we measure throughput in both prefill-heavy and decode-heavy settings across three different batch sizes. At a batch size of 1, parameter IO has a much greater impact on throughput compared to KV cache IO, resulting in slightly lower throughput for Block Transformer. However, as the model sizes increase beyond a certain point, the increased KV cache memory causes this trend to reverse. With a batch size of 32, our models achieve significantly higher throughput. To ensure that the improvements in decode-heavy settings are not solely due to gains in the prefill phase from not needing to forward the token decoder, we also experiment with a setting without a prompt. The results, summarized in Figure 11, show consistent performance improvements.

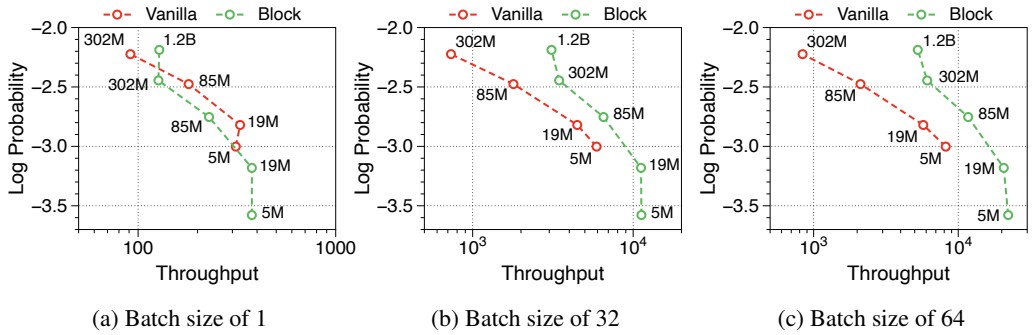

Figure 9: Pareto frontier of throughput to language modeling performance in the prefill-heavy setting. We set the input and output sequence lengths as 2048 and 128, respectively. The numbers denote the number of non embedding parameters in each model variants. We note that most vanilla models are out of memory from the batch size of 128.

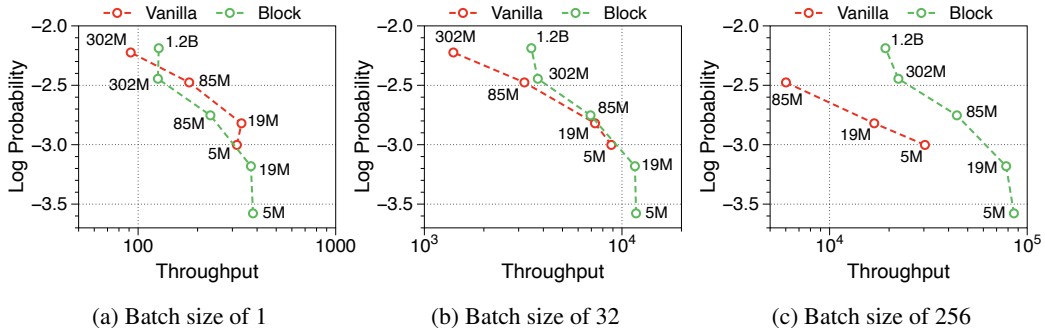

Figure 10: Pareto frontier of throughput to language modeling performance in the decode-heavy setting. We set the input and output sequence lenghts as 128 and 2048, respectively. In the batch size of 256, the vanilla model with the parameters of 302M is excluded due to out of memory issues.

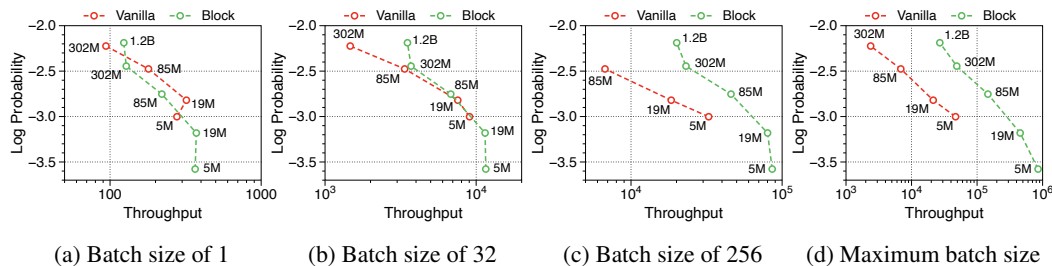

Figure 11: Pareto frontier of throughput without any input sequences. This setting is for the only decode phase, where the input and output sequence lengths are set to 1 and 2048, respectively. The numbers denote the number of non embedding parameters in each model variants.

Moreover, we compare the throughput of vanilla and Block Transformer models across various context lengths under two scenarios. In Figure 12, each point corresponds to the same order of model sizes. Our models demonstrate remarkable speed improvements, and even when the context length is increased by four or eight times, they outperform the vanilla models with a context length of 2K. By reducing the context length at the block decoder by a factor of block length, our models achieve faster generation speeds even with much longer context length.

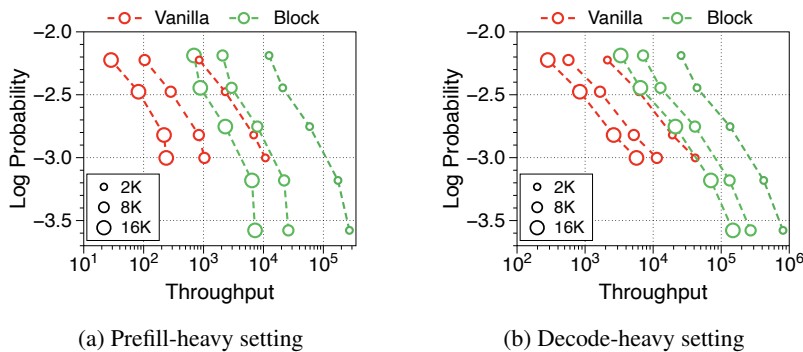

(a) Prefill-heavy setting             (b) Decode-heavy setting

Figure 12: Pareto frontier of throughput with varying context lengths. We set the prompt length to 128 in prefill-heavy scenarios and the output length to 128 in decode-heavy scenarios.

## K  Position-wise loss by parameter allocation ratio

We summarize the position-wise loss for three different model sizes in Figure 13. We confirm that changing the model size does not alter the overall trend, which exhibits a U-shape pattern depending on the token position. Additionally, we observe that a larger block decoder consistently improves the likelihood of earlier tokens, while a larger token decoder improves the likelihood of later tokens.

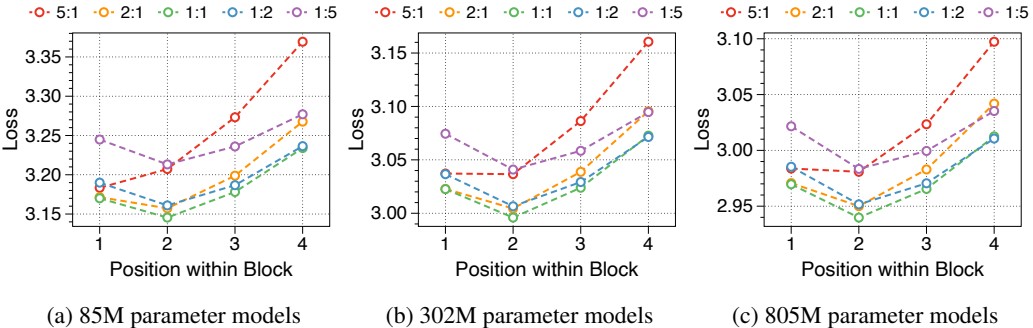

(a) 85M parameter models       (b) 302M parameter models       (c) 805M parameter models

Figure 13: Position-wise loss based on the model sizes and parameter allocation ratios. All models are trained on about 8 billion tokens with a block length of four. The parameter number indicates the sum of non-embedding parameters in block and token decoders, and the ratio represents the proportion of parameters between them.

## L  Loss trend by allocation ratio and block length

We analyze average loss in Figure 14 and position-wise loss in Figure 15 and Figure 16, adjusting for three block lengths and five allocation ratios across two model sizes. Surprisingly, all experimental results demonstrate the same trend. Notably, shorter block lengths favor larger block decoders, while longer block lengths benefit from larger token decoders. The rationale behind this trend becomes apparent through an examination of position-wise perplexity, particularly by observing the changes in loss for the first token and the variations in loss for later tokens. We believe that our extensive ablation studies will facilitate the determination of parameter ratios tailored to the specific scenarios for which the Block Transformer is designed.

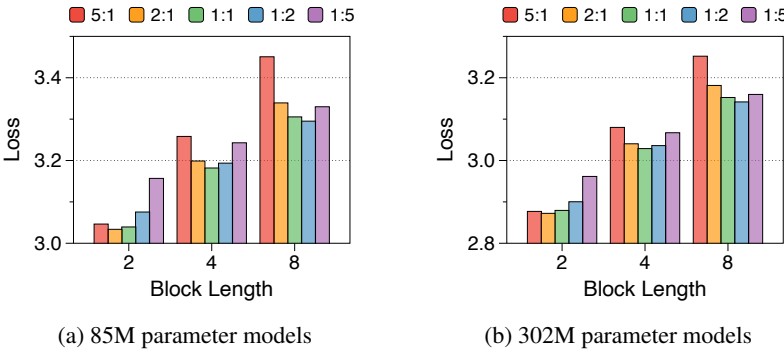

(a) 85M parameter models        (b) 302M parameter models

Figure 14: Loss by varying block lengths and the parameter allocation ratios. The numbers indicate the sum of non-embedding parameters in the block and token decoders.

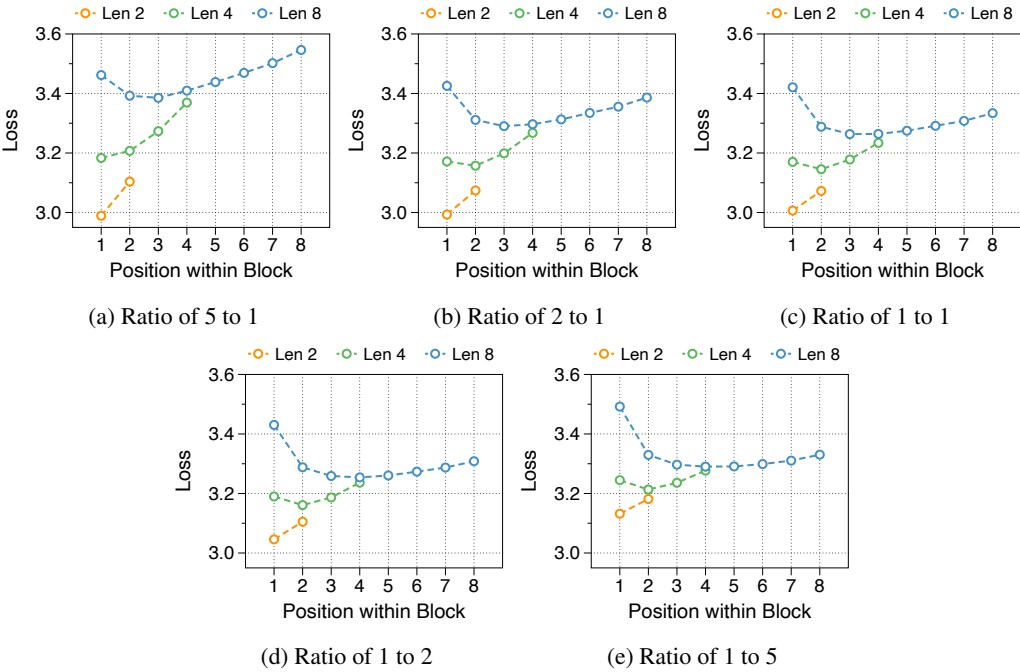

(a) Ratio of 5 to 1      (b) Ratio of 2 to 1      (c) Ratio of 1 to 1

(d) Ratio of 1 to 2      (e) Ratio of 1 to 5

Figure 15: Position-wise loss in relation to block length using three different parameter ratios. The models have 85M non-embedding parameters.

# M    Pareto frontier of throughput by allocation ratio and block length

While we have analyzed the optimal parameter ratio and block length from a perplexity perspective, we also evaluate which settings perform best from a throughput standpoint. The Pareto frontier for all model variants is depicted in Figure 17. Although there is a trade-off between throughput and performance, two clear findings emerge from the extensive combinations. First, the larger the token decoder, the higher the throughput improvement. Despite the token decoder consumes more FLOPs, the significantly shorter context length does not add overhead to the actual generation speed. Conversely, the block decoder, with its longer context length compared to the token decoder, hinders throughput as its size increases. The second observation is that longer block lengths significantly benefit throughput because they effectively reduce the context length. In conclusion, to optimize inference throughput, the token decoder should be enlarged, and the block length increased. However, to also consider perplexity, it is necessary to finely adjust the total model size, the allocation ratio, and the block length.

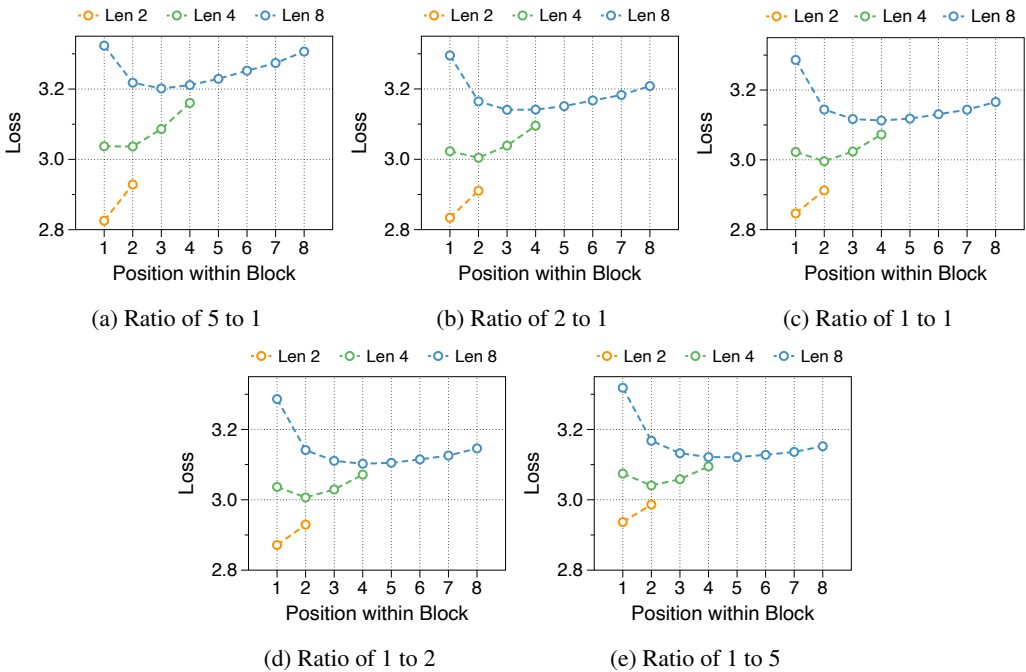

Figure 16: Position-wise loss in relation to block length using three different parameter ratios. The models have 302M non-embedding parameters.

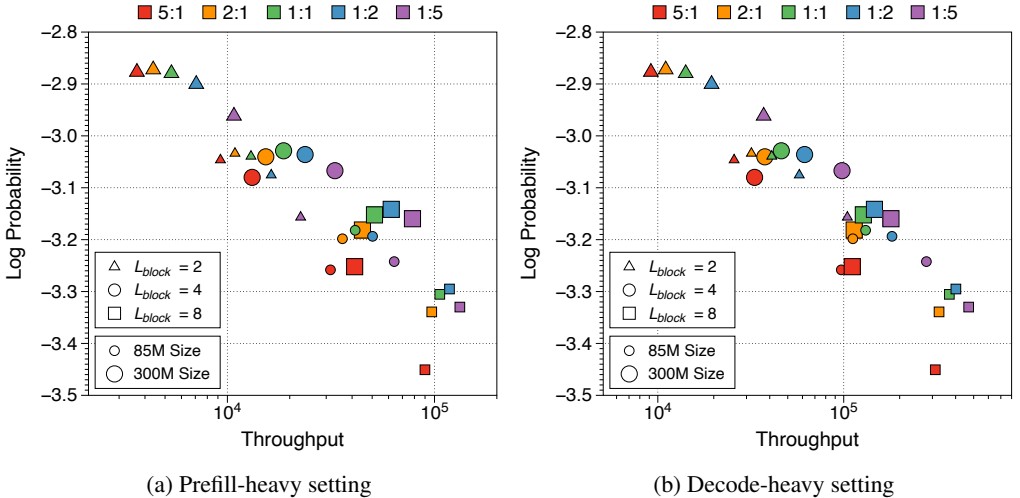

Figure 17: Pareto frontier of throughput to language modeling performance across various parameter allocation ratios, block lengths, and model sizes. Throughput is measured in the number of output tokens generated per second. The input and output sequence lengths are set to 2048 and 128 for the prefill-heavy setting, and 128 and 2048 for the decode-heavy setting. All model variants are trained on 8 billion tokens.

# N   Ablation studies on components of Block Transformer

## N.1   Embedder design

We compare three methodologies as embedder components in Figure 18. Surprisingly, the lookup strategy using an embedding table shows faster convergence than the transformer-based encoder, despite eventually reaching the same level of performance with prolonged training. Although increasing the number of layers of encoders could potentially improve performance, we choose not to pursue this due to its detrimental impact on inference throughput. Using a fixed number of CLS tokens allows for flexibility in adjusting the length of each block. Drawing inspiration from studies that adaptively allocate computational costs based on the difficulty of predictions [68, 4], this strategy could be effectively utilized when designing a Block Transformer capable of handling adaptive output lengths.

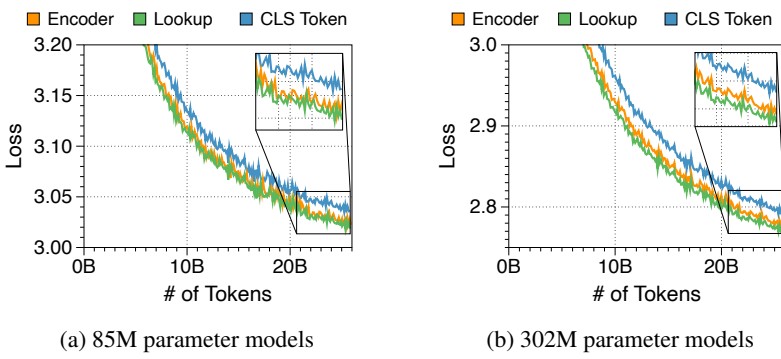

(a) 85M parameter models    (b) 302M parameter models

Figure 18: Training loss curve for three embedder components across two model sizes. We use a three layer RoBERTa model with a dimension of 256, and average the embeddings of three CLS tokens from the RoBERTa model.

## N.2   Token decoder design

In Figure 19, we compare three components for the optimal design of the token decoder. Prefix decoding outperform other strategies, particularly when the prefix length is increased, leading to a significant boost in performance. Given that the token decoder has a short context length, extending the prefix length does not substantially slow down the actual generation speed. However, since FLOPs increase proportionally, we set the prefix length to two as the main configuration to maintain a balance between performance and computational efficiency.

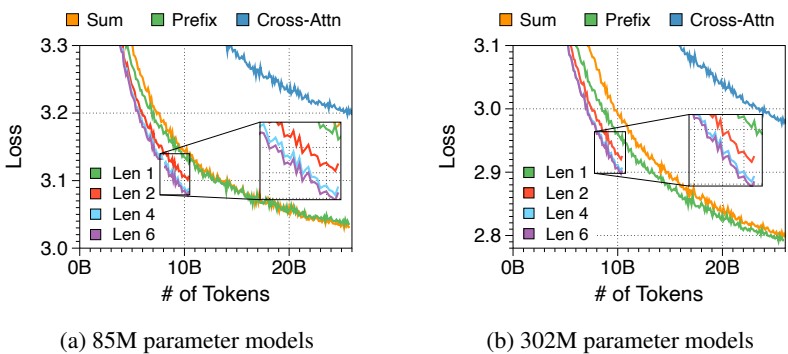

(a) 85M parameter models    (b) 302M parameter models

Figure 19: Training loss curve for three token decoder components across two models sizes. For the prefix method, we train the models with four different prefix lengths for block embeddings.

## O   Long-context modeling ability

**8K context length on the PG19 dataset**   To further support the effectiveness of our proposed block language modeling in capturing full context, we conducted experiments with an 8K context length (refer to Figure 20). However, due to limited computational resources, we pretrained only a 70M parameter vanilla model and a 170M parameter Block model. Following prior work [84, 82, 87] that used token position-wise perplexity on the PG19 dataset to demonstrate the utilization of global information in long contexts, we evaluated our Block Transformer in the same manner with 8K context length (refer to Figure 4b of the main paper for 2K context window). Even with a extended 8K context window, our models effectively utilized the full context, showing a decreasing loss trend as token position increased, similar to the vanilla model. Besides, consistent with Table 1 in the main paper, the 170M block model outperformed the 70M vanilla model in terms of perplexity.

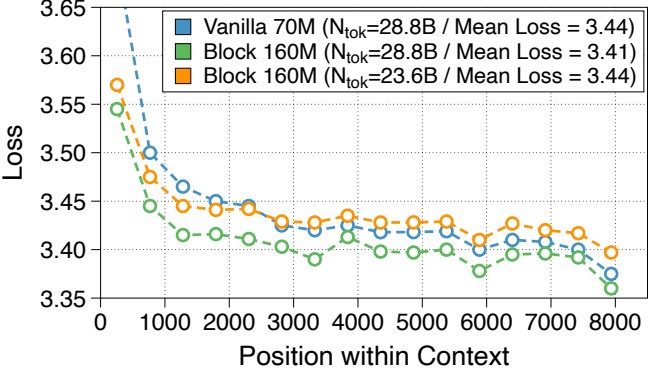

Figure 20: Long-context modeling ability up to 8K tokens. Average loss at different token positions (512-token bins) within 8192-token snippets from the entire PG19 test set. Blue and green lines show Vanilla and Block Transformers, trained on 28.8B tokens with context length of 8192. Orange line shows a vanilla-loss-equivalent Block Transformer checkpoint, at 23.6B tokens. Both architectures achieve lower loss as context length increases, with the Block Transformer performing relatively stronger at early tokens. Block Transformer achieves lower loss at all positions up to 8K.

**Performance on the Needle-In-a-Haystack task**   To precisely evaluate long-context modeling ability of LLMs, recent benchmarks, such as Needle-In-a-Haystack (NIAH) [39], LongBench [7], and ZeroScrolls [69], are typically utilized. However, to the best of our knowledge, these benchmarks mostly evaluate instruction-tuned models, as opposed to our pretrained base models. Nevertheless, we slightly modified the tailored prompt with an instruction version to evaluate the performance. Following prior work [67], we construct the context by first sampling 2K-length snippets from concatenated essays written by Paul Graham as the "haystack", and then inserting a "needle" containing key information in a random location. Following Reid et al. [67], we use this needle format: "`The special magic {city} number is:  {number}`." Here, {city} is a randomly chosen city name, and {number} is a random 7-digit number. We then append a prompt that queries to model to retrieve the 7-digit number. We consider two prompt formats:

1. **Gemini prompt**   Format is as follows: "`<context>\ncontext\n</context>\n\nWhat is the special magic {city} number?\n\nHere is the magic number from the context:`". We mostly followed the NIAH prompt used in Gemini, but we excluded the "Don't give information outside the document or repeat your findings" part, as our models are not instruction-tuned.

2. **Verbatim prompt**   Format is as follows: "`<context>\ncontext\n</context>\n\nquesti on\n\nThe special magic {city} number is:`". Here, we used the exact same format as that in the needle to query the model.

We measured the accuracy by generating 20 new tokens, and considering a prediction correct if the generated text contains the 7-digit number. Tables 5 and 6 presents a comparison of accuracy between vanilla and Block Transformers across two different prompts. We found that Block Transformers

perform equally or stronger than loss-equivalent vanilla models, consistently across (1) needle locations, (2) model scales and (3) prompt variants. These results confirm that the Block Transformer, like the vanilla models, can effectively retrieve global information contained within the 2K context length. With the Gemini prompt, we observed an accuracy trend that was very similar to the perplexity trend of the vanilla vs. block models. Near-perfect performance with the Verbatim prompt supports the long-sequence modeling capabilities of our models even when context information is squeeze into a single context embedding. We believe this parity between vanilla and Block Transformers on 2K context length will extend to 8K and beyond.

Table 5: Accuracy on Needle-In-a-Haystack task with Gemini prompt. Note that depth refers to the relative of the location of the needle within the haystack, in percentages.

| Models | N-Emb | Depth | | | | | | | | | | | Mean |
|---|---|---|---|---|---|---|---|---|---|---|---|---|---|
| | | 0 | 10 | 20 | 30 | 40 | 50 | 60 | 70 | 80 | 90 | 100 | |
| Vanilla | 19M | 0.00% | 0.00% | 0.00% | 0.00% | 0.00% | 0.00% | 0.00% | 0.00% | 0.20% | 0.80% | 6.40% | 0.67% |
| | 85M | 21.00% | 16.40% | 21.80% | 27.40% | 36.60% | 28.00% | 26.80% | 40.20% | 41.80% | 37.60% | 22.80% | 29.13% |
| | 300M | 46.20% | 69.00% | 72.80% | 78.60% | 76.40% | 70.40% | 71.80% | 74.80% | 73.80% | 78.40% | 66.20% | 70.76% |
| Block | 85M | 5.60% | 2.40% | 0.80% | 0.80% | 0.20% | 1.00% | 0.80% | 1.00% | 2.60% | 1.80% | 6.40% | 2.13% |
| | 300M | 23.40% | 52.60% | 52.60% | 46.60% | 46.00% | 49.20% | 58.40% | 70.40% | 64.00% | 53.60% | 18.40% | 48.65% |
| | 800M | 35.80% | 74.00% | 76.40% | 78.40% | 69.80% | 77.40% | 76.40% | 79.00% | 75.20% | 72.80% | 53.60% | 69.89% |
| | 1.2B | 57.20% | 86.60% | 88.80% | 85.60% | 80.40% | 85.20% | 90.40% | 89.20% | 91.00% | 90.40% | 78.80% | 83.96% |

Table 6: Accuracy on Needle-In-a-Haystack task with Verbatim prompt.

| Models | N-Emb | Depth | | | | | | | | | | | Mean |
|---|---|---|---|---|---|---|---|---|---|---|---|---|---|
| | | 0 | 10 | 20 | 30 | 40 | 50 | 60 | 70 | 80 | 90 | 100 | |
| Vanilla | 19M | 8.20% | 1.40% | 3.00% | 6.80% | 7.80% | 12.60% | 45.40% | 65.80% | 63.40% | 84.60% | 99.40% | 36.22% |
| | 85M | 95.60% | 99.40% | 99.00% | 99.40% | 99.20% | 99.20% | 99.00% | 99.60% | 99.60% | 99.00% | 95.60% | 98.60% |
| | 300M | 99.60% | 100.00% | 100.00% | 99.80% | 100.00% | 100.00% | 99.80% | 100.00% | 100.00% | 100.00% | 99.80% | 99.91% |
| Block | 85M | 96.20% | 97.60% | 96.20% | 96.60% | 98.40% | 98.00% | 97.20% | 98.80% | 99.00% | 99.40% | 96.20% | 97.60% |
| | 300M | 90.20% | 99.40% | 99.60% | 99.20% | 98.60% | 99.60% | 99.60% | 99.80% | 99.80% | 99.20% | 99.20% | 98.56% |
| | 800M | 95.20% | 99.40% | 98.80% | 98.80% | 98.80% | 98.80% | 99.00% | 99.40% | 99.20% | 97.40% | 99.60% | 98.58% |
| | 1.2B | 92.60% | 98.40% | 99.40% | 98.80% | 99.60% | 99.60% | 98.80% | 99.80% | 99.80% | 99.20% | 98.00% | 98.55% |

## P Uptraining strategy for training efficiency

Ainslie et al. [2] have demonstrated the significance of weight initialization for effectively uptraining models. Our extensive ablation studies reveal the optimal strategies for the Block Tramsformers: (1) Dividing a vanilla transformer layer in half and assigning each half to the block and token decoders, respectively, outperforms assigning the same weights of selected layers to both. (2) Initializing the input block embedding as the average of token embeddings within the block improves performance. (3) Initializing token decoder prefixes by replicating the context embedding enhances convergence. As depicted in Figure 21, these initialization techniques allow uptrained models to nearly match fully pretrained models. While larger models generally require longer uptraining, this approach still converges faster and recovers performance better than random initialization.

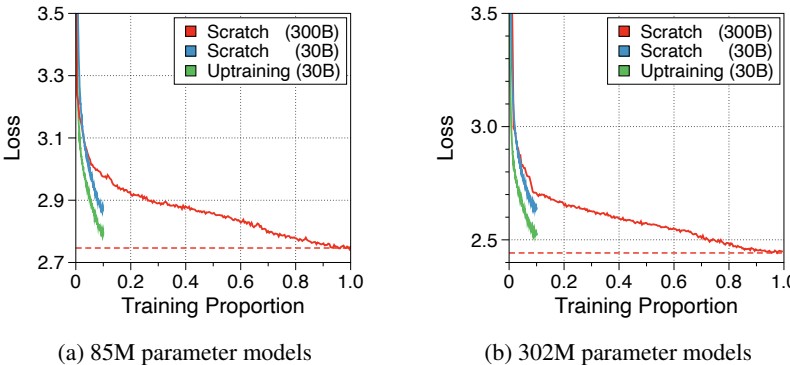

(a) 85M parameter models    (b) 302M parameter models

Figure 21: Training loss curve of uptraining strategy for two model sizes. Scratch denotes pretraining models from randomly initialized weights. The numbers in parentheses represents the number of training tokens.

## Q Performance comparison to MEGABYTE

MEGABYTE propose a global-to-local architecture similar to ours, but their emphasis on efficient training leads to different conclusions. For instance, they claim that a model structure with a block decoder six times larger than the token decoder is optimal, while overlooking the significance of local computation within the token decoder. However, our observations indicate that increasing the block decoder size is detrimental to throughput, and significantly reducing token decoder severely impacts language modeling performance. This is evident in Figure 22, where our reimplementation of MEGABYTE, based on their reported results, demonstrates considerably lower generation speed and performance than our baseline model in both prefill-heavy and decode-heavy settings. In light of this, we believe that our findings, focused on efficient inference, will open up new directions for global-to-local language models.

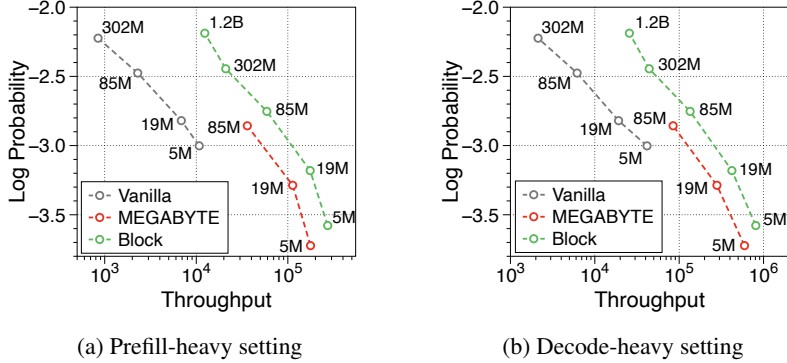

(a) Prefill-heavy setting    (b) Decode-heavy setting

Figure 22: Pareto frontier of throughput comparing our Block Transformer to MEGABTYE models. The numbers adjacent to each point indicte the number of non-mebedding parameters.

## R Visualization of attention scores in Block Transformer

We visualize the attention scores from both block and token decoder in Figure 23 and Figure 24. In block decoders, we observe a similar pattern of attention sinking to the first token. Previous research has taken advantage of this by keeping the first token as a global token to prevent performance drop when compressing long sequences of past tokens. We believe this approach could also benefit Block Transformers. Furthermore, the attention map in token decoders shows that later tokens strongly attend to the context embedding. This suggests that the global context is effectively compressed within them, which aligns with the insights in Section 4.1.

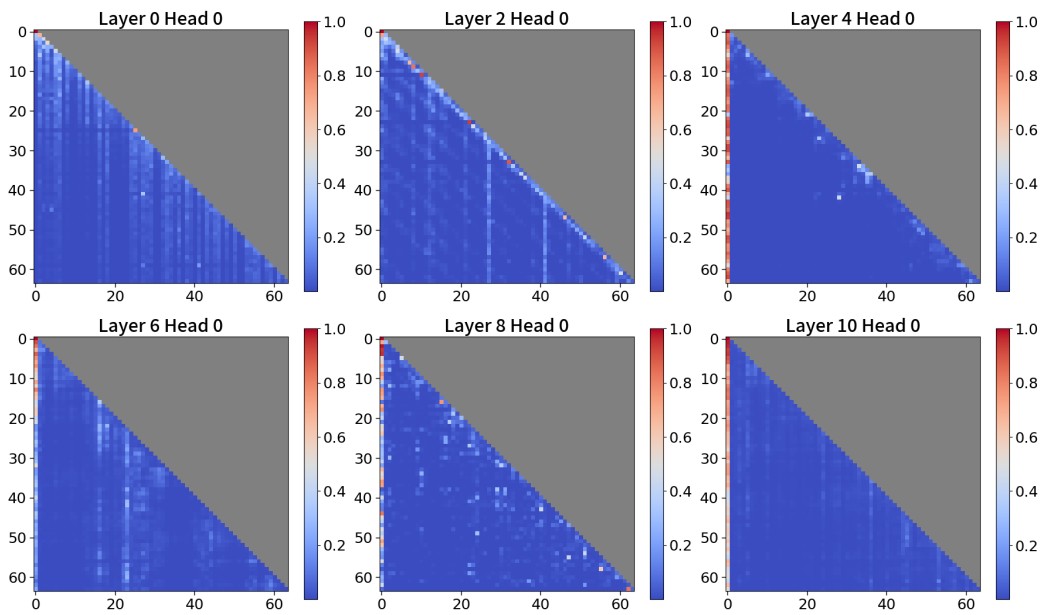

Figure 23: Visualization of attention scores in the block decoder. For clarity, we visualize only the first 64 sequences out of a total context length of 512. The causal mask parts are marked in gray.

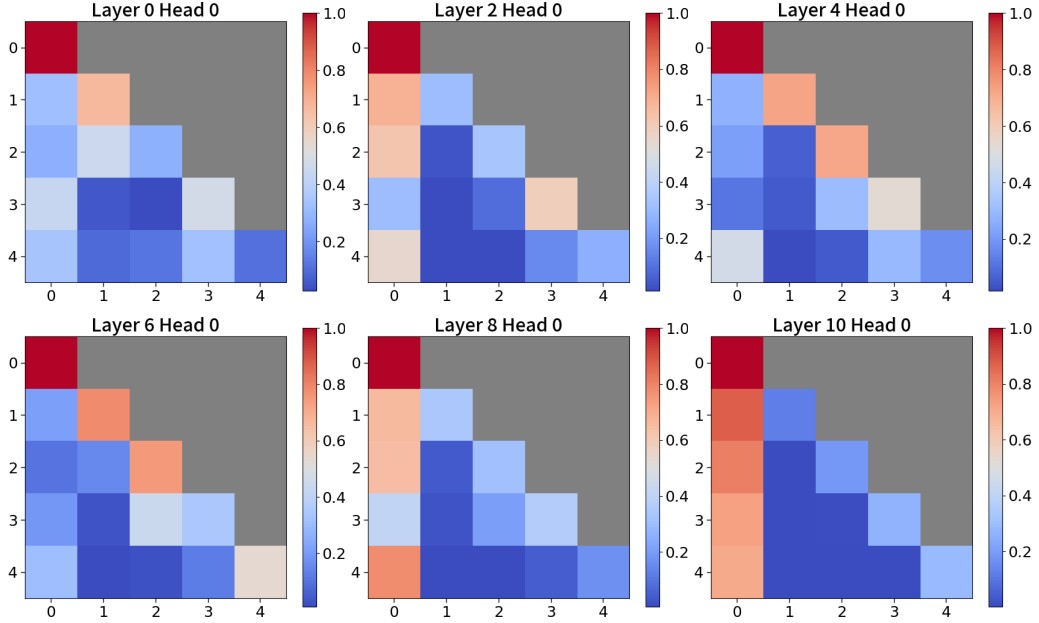

Figure 24: Visualization of attention scores in the token decoder. A total sequence length of attention scores is 5, since the block length is 4 and the prefix length is 2. The causal mask parts are marked in gray.

# S    Analysis on the context block embedding

To investigate whether global-to-local language modeling utilizes full context, we examine the information stored in context block embeddings. Specifically, given that the input token and context embedding share the same latent space in the token decoder, we analyze the three closest vocabulary terms to prefixes, which are projected from the context embedding, as shown in Table 7. We use a Block Transformer with 1.2 billion non-embedding parameters and prefix decoding with a prefix length of two. There are several interesting findings. The second prefix typically contains information about the last token of the current block. This suggests that the block decoder incorporates information about that specific token, rather than the previous sequences, to better predict the first token of the next block. Conversely, the first prefix of the context embedding contains uninterpretable tokens, indicating that it serves primarily to capture the global context as much as possible. This is further supported by Figure 24, which shows that later tokens in the token decoder tend to attend more to this prefix.

Table 7: Qualitative examples of the nearest token to the block embedding. We use a Block Transformer model with 1.2 billion non-embedding parameters. Utilizing prefix decoding with a length of two, we summarize the top three closest tokens for two positions of prefixes based on an embedding matrix from the token decoder. We randomly sample the input sequences from the Pile dataset.

| Sample | Tokens | Top-k | Block #0 | Block #1 | Block #2 | Block #3 | Block #4 |
|---|---|---|---|---|---|---|---|
| #0 | Input | - | \n\n#### Card | iff\n\n The | exuberant capital | of Wales, compact | Cardiff has recently |
| | Nearest | k=1 | ('<lendoftextl>', ' Card') | (' the', 'The') | (' guarantee', ' captial') | (' guranteee', ' compact') | (' the', ' has') |
| | | k=2 | ('the', 'Card') | ('<lendoftextl>', ' the') | ('ocardial', 'captial') | (' the', ',') | (',', ' recently') |
| | | k=3 | ('.', 'card') | ('219', ' The') | ('28', ' Capital') | (' unfamiliar', 'compact') | ('.', 've') |
| #1 | Input | - | the medieval Jewish community | , who were not | allowed to bury their | dead within the city | , would take bodies |
| | Nearest | k=1 | (' and', ' community') | (' the', ' not') | ('maybe', ' their') | (' LOSS', 'City') | (' deteriorated', ' body') |
| | | k=2 | (',', ' Community') | (' and', ' were') | (' LOSS', 'Their') | (' removed', ' City') | ('iding', ' bodies') |
| | | k=3 | (' the', 'community') | ('.', ' are') | (' and', ' Their') | ('otten', ' city') | ('pped', 'Body') |
| #2 | Input | - | to six daily | Fort William (£28 | .20, 3 | ¾ hours, four | to five daily), |
| | Nearest | k=1 | ('<lendoftextl>', ',') | (' fiercely', ' 28') | ('ijing', ' 3') | ('ulsions', ' four') | ('illes', ',') |
| | | k=2 | (' the', '),') | (' foe', '28') | ('\n     ', '3') | (' fierecely', ' 4') | ('yscall', '),') |
| | | k=3 | (' and', ']\\]') | ('illes', ' 30') | ('á¿', ' 4') | ('\n     ', ' three') | ('boats', '!),') |
| #3 | Input | - | can get almost anywhere | in Britain without having to drive.\n | \nThe main public | transport options are train |
| | Nearest | k=1 | ('<lendoftextl>', ' anywhere') | ('uin', ' having') | (' the', '.') | (' the', ' public') | ('onet', ' train') |
| | | k=2 | (' the', ' anything') | (' [...]', ' without') | (' and', 'Č') | ('.', ' Public') | ('stuff', 'train') |
| | | k=3 | ('.', 'anything') | (' the', ' have') | (',', '?).') | (' in', 'Public') | ('atisfaction', ' Train') |
| #4 | Input | - | \n\n**Length | ** : 2 miles | ; two to four | hours\n\nIt | 's fitting to start |
| | Nearest | k=1 | (' the', 'length') | (' the', ' miles') | (' the', ' four') | (' the', 'It') | (' the', ' start') |
| | | k=2 | ('<lendoftextl>', ' length') | (' and', 'km') | ('079', ' two') | (' in', ' It') | ('305', ' started') |
| | | k=3 | (' and', 'Length') | (' in', ' mile') | (' and', ' 4') | (' and', ' it') | (',', ' starts') |
| #5 | Input | - | the English church. | If this is the | only cathedral you visit | in England, you | 'll still walk away |
| | Nearest | k=1 | ('<lendoftextl>', '.') | (' the', ' the') | ('zione', ' visit') | ('zione', ' you') | ('aciones', ' away') |
| | | k=2 | (' and', '˘).') | ('cciÃ³n', 'The') | ('icions', ' visiting') | (' Heather', ' You') | (' 326', ' walk') |
| | | k=3 | (' the', ')$.') | (' and', ' this') | ('opsis', ' visits') | ('icions', 'You') | (' the', ' walked') |
| #6 | Input | - | \n\nStart at | the 1 **Store | y Arms car park | ** off the A | 470. A clear |
| | Nearest | k=1 | ('<lendoftextl>', ' at') | (' the', ' Store') | ('áĦÏ', ' Park') | (' and', ' A') | ('etus', ' clear') |
| | | k=2 | (' the', 'At') | ('áĦÏ', 'Store') | (' and', 'Park') | (' the', 'A') | (' the', ' Clear') |
| | | k=3 | (',', ' At') | (' and', ' store') | ('ishops', 'park') | ('.', ' a') | ('Ã§Ã£o', 'Clear') |

