# OpenReview forum: "Block Transformer: Global-to-Local Language Modeling for Fast Inference"
_NeurIPS.cc/2024/Conference — NeurIPS 2024 poster_

### Official Review · Reviewer_BaSr · 2024-06-30

**Soundness:** 3
**Presentation:** 3
**Contribution:** 3
**Rating:** 6
**Confidence:** 5

**Summary:**

This paper proposes a Block Transformer architecture which adopts hierarchical global-to-local modeling to mitigate the computational cost and KV cache memory of Self Attention. Block Transformer isolate the global modeling with three blocks: Embedder, Block Decoder, and Token Decoder. Embedder encodes block information for auto-regressive modeling in Block Decoder. The aggregated information is plugged into Token Decoder for final token-level decoding. Block Transformer shows 10-20x inference improvement under the similar perplexity than traditional global Transformer. This paper gives a detailed ablation about the architecture design of Block Transformer. Besides, the global modeling capability and uptraining strategy is also discussed.

**Strengths:**

1. The paper is well organized and the experiments are solid and comprehensive.
2. Block Transformer trades inference efficiency with model parameters, which is a bold innovation.
3. Block Transformer achieves low-cost inference compared with standard Transformer.
4. The architecture analysis and ablation studies show the effectiveness of Block Transformer design.

**Weaknesses:**

1. Block Transformer uses much bigger model size to compensate for the performance loss. There are countless problem for that, including training partitioning and inference infrastructure.
2. The actual inference efficiency comparison is doubtful. For example, in Table 2, The Decode throughput of Block Transformer is much bigger than standard Transformer under same model size. Since Block Transformer only saves Attention computation and the FFN computation stays the same, I'm confused that why attention computation occupies almost most of the overall computation.
3. The long-sequence modeling capability is not evaluated. Since Block Transformer squeezes the context representation, it is questionable that if Block Transformer can retrieve the global context information. I think some common long-sequence experiments will help, e.g., Needle-in-a-Haystack.
4. The scaling property is not discussed good enough. For example, in Figure 2, there are some results in different model sizes. However, there is not a "scaling law" for Block Transformer. Besides, the scaling does not look promising with my human eye in Figure 2.

**Questions:**

1. Concerns in the Weaknesses part.
2. I'm curious about the inference experiment setting. Are FlashAttention and FlashDecoding techniques used for vanilla Transformer or Block Transformer? I believe it is already a necessary part in 2024 year.

**Limitations:**

The limitation is discussed.

---

> ### Author Rebuttal · Authors · 2024-08-07
>
> We thank you for the comments, and acknowledging the innovation of trading inference efficiency and parameters, the efficiency of Block Transformer, solid experiments, and good paper organization. We address weaknesses below.
>
> .
>
> **W1. Bigger model size**
>
> Despite bigger model size, Block Transformers achieve **higher throughput** and **lower total memory usage** during inference. For training, it is possible to **uptrain existing vanilla models** with ~10% of training steps, and training costs can be **cheaper under constrained inference budget.**
>
> **Inference case**
>
> - **Total memory usage**: Block Transformers can achieve significantly higher batch sizes under same hardware, as the KV cache **memory usage per sample of Block Transformers is 3—6x times lower**, despite the larger size (`Table 2`). Note that the KV cache of a single sample can be as large as the parameters of the entire model (`Table 2`).
> - **Throughput**: our main results show significant gains in throughput over vanilla models *under same hardware constraints*.
> - These advantages extend into the multi-GPU inference scenario. Tensor parallelism can further reduce per-GPU model parameter memory and leaves more room for more KV cache, which is advantageous for the Block Transformer.
>
> **Training case**
>
> - IsoFLOP analysis in `Section 3.6` shows that Block Transformers can achieve better performance **and** significantly higher throughput using the **same training FLOPs** as vanilla models when constrained by inference budget (we discuss further in W4).
> - `Section 3.7` shows that it is possible to **uptrain existing vanilla models** into Block Transformers, significantly reducing development costs. Uptrained models approach the performance of model pre-trained from scratch, using just `10%` of training steps (`Figure 5a` and `Appendix M`).
>
> .
>
> **W2. Doubts on inference efficiency comparison**
>
> We point out that **Block Transformer does not “only save attention computation”**, but have a wide range of advantages stemming from our architecture design. Please refer to `Section 2` and `Appendix D`. We also provide a **visual recap of advantages in `Figure 1` of the attached PDF.**
>
> We re-iterate four key advantage of our architecture:
>
> 1. `Attention + FFN` Token decoder does not need to prefill prompt tokens.
> 2. `Attention` KV cache IO at the token decoder is reduced by $L/L_B=2048/4=256\times$.
> 3. `Attention + FFN` Block decoder operates at the block level, reducing overall costs by $L_B=4$. KV cache IO cost is reduced quadratically.
> 4. `Attention + FFN` Overall KV cache storage is reduced by $L_B=4\times$ in the block decoder and nearly eliminated in the token decoder. This enables higher batch sizes and thus higher compute utilization.
>
> We also empirically pinpoint these benefits in **new detailed measurements in `Table 1`** of the `attached PDF`. These show that Block Transformer speeds up all segments of computation relative to a loss-comparable vanilla model: {attention, FFN} operations at {lower, upper} layers during {prefill, decoding} stages under {prefill, decode}-heavy settings.
>
> .
>
> **W3. No evaluation of long-sequence modeling capability**
>
> `Figure 4b` of `Section 3.5` demonstrates the long-context modeling capabilities of Block Transformers. Our models achieve lower loss by utilizing more context up to 2K tokens on the PG19 test dataset. This is a standard benchmark used to evaluate long-context modeling capability [1, 2].
>
> We also extend this to 8K context in `Figure 2 of attached PDF`, with new models pretrained on 8K context for ~30B tokens. We show that the Block Transformer (1.2B) achieves lower loss at all token positions relative to a comparable vanilla baseline (300M), despite achieving significantly higher throughput (`Appendix Figure 10`).
>
> .
>
> **W4. Scaling property**
>
> In Appendix A, we discuss that rigorous Chinchilla-style scaling study is infeasible as it requires repeated training runs with appropriate LR schedules for each FLOP budget [3].
>
> However, expanding on W1, we do discuss in `Section 3.6` why our isoFLOP analysis (which is in the overtraining regime for the vanilla model) is relevant under current trends—where small models are significantly overtrained to maximize performance at a given inference budget [4]. I.e., to achieve maximize performance using plentiful training FLOPs under a tight inference budget with vanilla models, we must resort to training a small model into a highly suboptimal training regime. Since Block Transformers have significantly smaller inference costs, we can use larger models which can achieve **higher better performance given the same training FLOPs**, **under inference budget constraints**. This is what is shown in `Figure 4c`.
>
> .
>
> **Q2. FlashAttention and FlashDecoding**
>
> **Existing experiments**: all models in our paper are based on GPTNeoX using MHA on Huggingface and used eager attention implementation for inference, due to limited support for FlashAttention2 during the time of submission*.
>
> **FlashDecoding [5]**: New measurements using FlashAttention2’s FlashDecoding kernel* show a **similar pareto frontier as our existing results** in `Figure 3 of attached PDF`. Please compare with `Figure 2 of main paper`. We find that FlashAttention2 achieves `8-41%` throughput gains in vanilla models and `+8% to +31%` in block models, in the prefill-heavy setting. Gains are diminished in larger models. We observe `+16% to -37%` speedup across models in the decode-heavy setting, without a coherent trend.
>
> .
>
> **References**
>
> [1] Zhang, Zhenyu, et al. "H2o: Heavy-hitter oracle for efficient generative inference of large language models."
>
> [2] Xiao, Guangxuan, et al. "Efficient streaming language models with attention sinks.”
>
> [3] Hoffmann, Jordan, et al. "Training compute-optimal large language models.”
>
> [4] Touvron, Hugo, et al. "Llama: Open and efficient foundation language models.”

---

> ### Comment · Reviewer_BaSr · 2024-08-09
> **Response to Authors' Rebuttal**
>
> I appreciate authors' thorough and patient response. I have a better understanding of the experiment setting. However, I still have some of my previous concerns:
>
> 1. Measuring the perplexity on PG19 to evaluate the long-sequence modeling capability is still not a good experiment. Since some subsequent works show that H2O and Attention Sink are not good at some long context tasks, including Needle-in-a-Haystack, LongBench, and ZeroScrolls, I don't agree it's a **standard benchmark**.
>
> I believe it is a novel and insightful work. But there is still space for improvement. In a nutshell, the paper will look much stronger if the experiment is on a "**modern**" setting, including the latest evaluation settings and kernel techniques. I will keep my initial score.

---

> > ### Author Response · Authors · 2024-08-10
> >
> > Thank you for acknowledging our comments.
> >
> > We are glad that we have resolved many of your concerns including model size, inference efficiency, and scaling properties.
> >
> > While we have shown positive long context perplexity results up to 8K, we will also make sure to perform additional evaluation on more recent long context tasks.
> >
> > Regarding modern implementation, we would like to re-iterate that **Vanilla and Block Transformers both benefit from modern FlashDecoding kernels** (`Figure 3 of attached PDF`), maintaining our pareto results. Detailed measurements in `Table 2 of attached PDF` suggests that Block Transformers will still be faster overall, when orthogonally applying modern attention schemes such as MQA/GQA. We will also include these analyses and more in our final paper.

---

> > ### Author Response · Authors · 2024-08-13
> > **Experimental results on the Needle-in-a-Haystack task**
> >
> > We appreciate again the acknowledgement for the novelty and insightful contributions of our work.  To the best of our knowledge, recent long-context benchmarks like Needle-in-a-Haystack (NIAH), LongBench, and ZeroScrolls typically evaluate instruction-tuned models, as opposed to our pre-trained base models.  Nevertheless, we are pleased to share **additional results on the Needle-in-a-Haystack task**.
> >
> > We found that **Block Transformers perform equally or stronger than loss-equivalent vanilla models**, consistently across **(1) needle locations**, **(2) model scales** and **(3) prompt variants**.
> >
> > .
> >
> > ## Experimental settings.
> >
> > Following prior work [1], we construct the context by first sampling 2K-length snippets from concatenated essays written by Paul Graham as the “haystack”, and then inserting a “needle” containing key information in a random location. Following [1], we use this needle format: `The special magic {city} number is: {number}`.
> >
> > - `{city}` is a randomly chosen city name
> > - `{number}` is a random 7-digit number.
> >
> > We then append a prompt that queries to model to retrieve the 7-digit number. We consider two prompt formats:
> >
> > **1. Gemini prompt**
> >
> > Format: `<context>\n{context}\n</context>\n\nWhat is the special magic {city} number?\n\nHere is the magic number from the context:`
> >
> > We mostly followed the NIAH prompt used in Gemini [1], but we excluded the “Don’t give information outside the document or repeat your findings” part, as our models are not instruction-tuned.
> >
> > **2. Verbatim prompt**
> >
> > Format: `<context>\n{context}\n</context>\n\n{question}\n\nThe special magic {city} number is:`.
> >
> > Here, we used the exact same format as that in the needle to query the model.
> >
> > We measured the accuracy by generating 20 new tokens, and considering a prediction correct if the generated text contains the 7-digit number.
> >
> > .
> >
> > ## Experimental results.
> >
> > Note that depth refers to the relative of the location of the needle within the haystack, in percentages.
> >
> > **Gemini prompt**
> >
> > | Depth | 0 | 10 | 20 | 30 | 40 | 50 | 60 | 70 | 80 | 90 | 100 | Mean |
> > | --- | --- | --- | --- | --- | --- | --- | --- | --- | --- | --- | --- | --- |
> > | Vanilla 19M | 0.00% | 0.00% | 0.00% | 0.00% | 0.00% | 0.00% | 0.00% | 0.00% | 0.20% | 0.80% | 6.40% | 0.67% |
> > | Vanilla 85M | 21.00% | 16.40% | 21.80% | 27.40% | 36.60% | 28.00% | 26.80% | 40.20% | 41.80% | 37.60% | 22.80% | 29.13% |
> > | Vanilla 300M | 46.20% | 69.00% | 72.80% | 78.60% | 76.40% | 70.40% | 71.80% | 74.80% | 73.80% | 78.40% | 66.20% | 70.76% |
> > | Block 85M | 5.60% | 2.40% | 0.80% | 0.80% | 0.20% | 1.00% | 0.80% | 1.00% | 2.60% | 1.80% | 6.40% | 2.13% |
> > | Block 300M | 23.40% | 52.60% | 52.60% | 46.60% | 46.00% | 49.20% | 58.40% | 70.40% | 64.00% | 53.60% | 18.40% | 48.65% |
> > | Block 800M | 35.80% | 74.00% | 76.40% | 78.40% | 69.80% | 77.40% | 76.40% | 79.00% | 75.20% | 72.80% | 53.60% | 69.89% |
> > | Block 1.2B | 57.20% | 86.60% | 88.80% | 85.60% | 80.40% | 85.20% | 90.40% | 89.20% | 91.00% | 90.40% | 78.80% | 83.96% |
> >
> > **Verbatim prompt**
> >
> > | Depth | 0 | 10 | 20 | 30 | 40 | 50 | 60 | 70 | 80 | 90 | 100 | Mean |
> > | --- | --- | --- | --- | --- | --- | --- | --- | --- | --- | --- | --- | --- |
> > | Vanilla 19M | 8.20% | 1.40% | 3.00% | 6.80% | 7.80% | 12.60% | 45.40% | 65.80% | 63.40% | 84.60% | 99.40% | 36.22% |
> > | Vanilla 85M | 95.60% | 99.40% | 99.00% | 99.40% | 99.20% | 99.20% | 99.00% | 99.60% | 99.60% | 99.00% | 95.60% | 98.60% |
> > | Vanilla 300M | 99.60% | 100.00% | 100.00% | 99.80% | 100.00% | 100.00% | 99.80% | 100.00% | 100.00% | 100.00% | 99.80% | 99.91% |
> > | Block 85M | 96.20% | 97.60% | 96.20% | 96.60% | 98.40% | 98.00% | 97.20% | 98.80% | 99.00% | 99.40% | 96.20% | 97.60% |
> > | Block 300M | 90.20% | 99.40% | 99.60% | 99.20% | 98.60% | 99.60% | 99.60% | 99.80% | 99.80% | 99.20% | 99.20% | 98.56% |
> > | Block 800M | 95.20% | 99.40% | 98.80% | 98.80% | 98.80% | 98.80% | 99.00% | 99.40% | 99.20% | 97.40% | 99.60% | 98.58% |
> > | Block 1.2B | 92.60% | 98.40% | 99.40% | 98.80% | 99.60% | 99.60% | 98.80% | 99.80% | 99.80% | 99.20% | 98.00% | 98.55% |
> >
> > .
> >
> > These results confirm that the Block Transformer, like the vanilla models, can effectively retrieve global information contained within the 2K context length. With the Gemini prompt, we observed an accuracy trend that was very similar to the perplexity trend of the vanilla vs block models. Near-perfect performance with the Verbatim prompt supports the long-sequence modeling capabilities of our models even when context information is squeeze into a single embedding. We believe this parity between Vanilla and Block Transformers on 2K context length will extend to 8K and beyond.
> >
> > .
> >
> > We would appreciate it if you could reflect our additional results on FlashDecoding (modern implementation) and NIAH evaluation (modern evaluation) in your final score, as we believe these have adequately addressed your concerns.
> >
> > .
> >
> > [1] Gemini Team, Google. “Gemini 1.5: Unlocking multimodal understanding across millions of tokens of context.”

---

> ### Comment · Reviewer_BaSr · 2024-08-14
>
> I'm pleased to see that Block Transformer can effectively retrieve global information contained within the 2K context length, that is a strong indicator for long-context capability. I will increase my score to 6.

---

### Official Review · Reviewer_769s · 2024-07-10

**Soundness:** 3
**Presentation:** 3
**Contribution:** 2
**Rating:** 6
**Confidence:** 4

**Summary:**

The authors introduced the Block Transformer architecture to address the self-attention bottleneck. This is achieved by grouping input tokens into fixed-size blocks and applying self-attention at a corser level throughout the model. At the output layer, a token decoder predicts individual tokens from the block. The authors found that this hierarchical global-to-local modeling approach results in 10 to 20 times faster inference compared to vanilla transformers with similar perplexity.

**Strengths:**

- The topic of improving the efficiency of LLMs and making them more affordable is crucial and timely. Furthermore, as sequences scale, the self-attention bottleneck increases.
- The authors conducted extensive experiments and ablations with a modern setup. The evaluation includes perplexities and zero-shot downstream tasks. The models were trained for a significant number of tokens (300B, which is more than an epoch on the Pile), making the result more trustworthy.
- The paper is well-written and easy to follow.

**Weaknesses:**

- The concept appears similar to that of the Funnel Transformer [1], with the main exception that the aggregation happens only once at the token level.
- The Pareto front of the decoding throughput is only improved with batch sizes greater than 32, which may not often be the case.
- The higher inference throughput comes at the cost of more training compute and memory. The proposed methods perform worse than the vanilla model at an equivalent size.

[1] Dai, Zihang, et al. "Funnel-transformer: Filtering out sequential redundancy for efficient language processing." Advances in neural information processing systems 33 (2020): 4271-4282.

**Questions:**

- Is the baseline using FlashAttention2 [2] and MQA/GQA [3] ? These two modifications have become standard and significantly reduce the bottleneck of the attention.
- Could you explain the differences between your methods and the Funnel Transformer?
- Given the Funnel Transformer and the many other sparse attention, do you maintain your claim that you "are the first to recognize the central role and inference-time benefits of both global and local modeling in autoregressive transformers, particularly the significance of local modules"?

[2] Dao, Tri. "Flashattention-2: Faster attention with better parallelism and work partitioning." arXiv preprint arXiv:2307.08691 (2023).
[3] Ainslie, Joshua, et al. "Gqa: Training generalized multi-query transformer models from multi-head checkpoints." arXiv preprint arXiv:2305.13245 (2023).

**Limitations:**

Yes.

---

> ### Author Rebuttal · Authors · 2024-08-07
>
> We thank you for the comments, and we are encouraged that you pointed out the trustworthiness of our extensive experiments with modern setup, and easy-to-follow writing. We address weaknesses below.
>
> **W1. Difference between Block Transformer and Funnel Transformers**
>
> **Major difference in key aspect—local attention**: we respectfully disagree that our Block Transformer is similar to existing pooling transformers, including Funnel Transformer [1], Hourglass [2], and CANINE [3]. Our approach is fundamentally different, as it applies **local attention** in the token decoder instead of maintaining global attention with pooling and up-sampling.
>
> Pooling transformers achieve speedup by pooling as the depth increases and then upsampling in the upper layers. However, they maintain global attention throughout all layers. In contrast, while global-to-local modeling applies a pooling operation at the token level, a primary feature is the locality of attention in the token decoder (upper layers). The token decoder applies attention only within the local window, leading to significantly higher compute utilization.
>
> We will include this discussion in our related works section.
>
> Refer to advantages of this local attention (and more) in our visual recap in `Figure 1 of attached PDF` , and details in `Section 2.4` and `Appendix D.2` . We also empirically pinpoint the benefits of the locality aspect (`Table 1 of attached PDF`), accounting for `~90% reduction` in walltime at the upper layers
>
> .
>
> **W2. Pareto-Frontier of decoding throughput is only improved with batch sizes greater than 32**
>
> We respectfully disagree, and would like to emphasize that trends in `Figures 7, 8` show the throughput gains increases as the model size increases, even at smaller batch sizes
>
> In fact, **Block 1.2B already surpasses Vanilla 300M** in both prefill-heavy and decode-heavy throughput **at batch size 1**. This is because KV cache saturates GPU memory even at small batch sizes, with larger models.
>
> Below, we show the throughput of models above 1B parameter with batch size 1 and 2. The advantage of Block 1.2B against the loss-equivalent Vanilla 300M model widens as batch size increases. We expect that Block 6.4B will perform between Vanilla 1.2B and 2.5B, and find that Block 6.4B is faster than Vanilla 1.2B. The gap also widens as batch size increases.
>
> | Batch Size 1 | Vanilla 85M | Block 300M | Vanilla 300M | Block 1.2B | Vanilla 1.2B | Vanilla 2.5B | Block 6.4B |
> | --- | --- | --- | --- | --- | --- | --- | --- |
> | Prefill Heavy | 216.26 | 158.34 | 115.24 | 154.37 | 113.05 | 79.63 | 110.64 |
> | Decode Heavy | 222.22 | 155.38 | 113.23 | 153.44 | 114.35 | 80.54 | 110.46 |
>
> | Batch Size 2 | Vanilla 85M | Block 300M | Vanilla 300M | Block 1.2B | Vanilla 1.2B | Vanilla 2.5B | Block 6.4B |
> | --- | --- | --- | --- | --- | --- | --- | --- |
> | Prefill Heavy | 422.18 | 316.31 | 224.51 | 290.84 | 203.49 | 152.86 | 211.27 |
> | Decode Heavy | 426.24 | 306.22 | 221.36 | 290.72 | 213.88 | 155.64 | 221.84 |
>
> .
>
> **W3. Block Transformer requires larger model size, and costs more training compute and memory**
>
> To mitigate model development costs (including training), it is possible to **uptrain existing vanilla models** with just `~10%` of training steps. We also find that training costs can be **cheaper under constrained inference budget.**
>
> - Uptraining existing vanilla models into Block Transformers approaches the performance of models pre-trained from scratch, using just 10% of training steps (`Section 3.7`, `Figure 5a`, `Appendix M`).
> - IsoFLOP analysis under in `Section 3.6` shows that Block Transformer can achieve **better performance** and **significantly higher throughput** using the **same training FLOPs** as vanilla models. Note that this assumes overtraining to fit a fixed budget constraint, in line with recent trends set by models such as Gemma and Llama [4] (`Section 3.6`).
>
> .
>
> **Q1. Do baselines use FlashAttention2 and MQA/GQA?**
>
> **Existing experiments**: all models in our paper are based on GPTNeoX using MHA on Huggingface and used eager attention implementation for inference, due to lack of support for FlashDecoding during the time of submission.
>
> **FlashDecoding [5]**: New measurements using FlashAttention2’s FlashDecoding kernel* show a **similar pareto frontier as our existing results** in `Figure 3 of attached PDF`. Please compare with `Figure 2 of main paper`. We find that FlashAttention2 achieves `8-41%` throughput gains in vanilla models and `+8% to +31%` in block models, in the prefill-heavy setting. Gains are diminished in larger models. We observe `+16% to -37%` speedup across models in the decode-heavy setting, without a coherent trend.
>
> **GQA**: We expect that both Vanilla and Block Transformers will benefit from GQA, in terms of the attention operations.
>
> - Q. what if FFN becomes the deciding factor after applying GQA?
> - A. new measurements show that Block Transformer is significantly faster than vanilla in **both attention and FFN operations** (`Table 1 in attached PDF`).
>
> **Q3. Maintaining our claim**
>
> We made this claim because we were the first to identify the locality in the token decoder as the source of significant inference-time benefits, in the context of global-to-local modeling (coarse global attention in lower layers and fine local attention in upper layers).
>
> We will revise the sentence to more clearly emphasize our recognition of the benefits of the local module in global-to-local modeling.
>
> .
>
> **References**
>
> [1] Dai, Zihang, et al. "Funnel-transformer: Filtering out sequential redundancy for efficient language processing."
>
> [2] Nawrot, Piotr, et al. "Hierarchical transformers are more efficient language models."
>
> [3] Clark, Jonathan, et al. “CANINE: Pre-training an Efficient Tokenization-Free Encoder for Language Representation.”
>
> [4] Touvron, Hugo, et al. "Llama: Open and efficient foundation language models.”
>
> [5] Dao, Tri, et al. "Flash-Decoding for long-context inference”

---

> > ### Comment · Reviewer_769s · 2024-08-09
> >
> > Thank you for the detailed rebuttal. Overall, I am pleased with the answers and clarifications provided, and I have adjusted my score accordingly.
> >
> > **W1.** I appreciate the clarification on the key difference between the proposed architecture and Pooling Transformers that is the locality in the decoding layers. I agree that local attention will significantly reduce computational requirements, particularly for long sequences.
> >
> > **W2.** I am satisfied with the provided numbers and suggest adding the table to the appendix.
> >
> > Figs. 7a and 8a do not show a clear Pareto front improvement at a batch size of 1, in contrast to Figs. 7b and 8b at a batch size of 32. I recommend extending these figures to include larger models to highlight the improvement.
> >
> > Additionally, I suggest revising the statement Lines 746-749: "At a batch size of 1, parameter IO has a much greater impact on throughput compared to KV cache IO, resulting in slightly lower throughput for block model. However, as the model sizes increase beyond a certain point, the increased KV cache memory causes this trend to reverse." Specifying that this trend reverses between 300M and 1.2B parameters may be helpful.
> >
> > **W3.** I am satisfied with the response.
> >
> > **Q1.** Thank you for the additional experiments on FlashAttention and GQA.
> >
> > **Q3.** Based on your response to **Q1**, I now have a better understanding of your claim. I appreciate your commitment to clarifying the sentence.

---

> > > ### Author Response · Authors · 2024-08-09
> > >
> > > Thank you for acknowledging our comments, and providing additional detailed feedback on our manuscript.
> > >
> > > We are glad that we could clarify the novelty and contribution of our work, and further show the generality of our results to practical settings and state-of-the-art implementations.
> > >
> > > We will include the additional analysis and clarifications in our final paper, with further results on more batch sizes and model sizes.

---

### Official Review · Reviewer_unsu · 2024-07-11

**Soundness:** 3
**Presentation:** 3
**Contribution:** 1
**Rating:** 5
**Confidence:** 2

**Summary:**

The paper introduces the Block Transformer architecture, which aims to improve inference speed in autoregressive language models by adopting a hierarchical global-to-local approach. The architecture separates the global context modeling into lower layers and local detailed interactions into upper layers, thus reducing the self-attention bottleneck. The authors demonstrate significant improvements in inference throughput without compromising perplexity.

**Strengths:**

1. The experiments in this paper are extensive, covering a variety of model parameters.
2. The proposed Block Transformer demonstrates improvements in computational efficiency, which is crucial for scaling up to longer sequences.

**Weaknesses:**

**Major Weakness**
As I understand it (please correct me if I'm wrong), the primary difference between Block Transformer and MEGABYTE [1] is whether the input is a token or a byte. The architecture of Block Transformer is nearly identical to that of MEGABYTE, which significantly limits the novelty and contribution of this work.

[1] MEGABYTE: Modeling Million-byte Sequences with Multiscale Transformers

**Questions:**

None

**Limitations:**

As discussed above.

---

> ### Author Rebuttal · Authors · 2024-08-07
>
> We thank you for the comments, and we are encouraged that you pointed out the improved efficiency of our Block Transformers with extensive experiments. The weakness of our paper is discussed as follows.
>
> **W1. Difference between Block Transformer and MEGABYTE.**
>
> We have discussed the main differences to previous works related to global-to-local modeling in `Appendix C.1`. **We believe our contributions are novel to Megabyte. Here’s why:**
>
> 1.  **Firstly, the primary goal of our proposed architecture, which utilizes global-to-local language modeling, is clearly different.** While MEGABYTE (including most hierarchical transformer works) focuses on **efficient pretraining**, we mainly focuses on **efficient inference**. MEGABYTE aimed to reduce training time by minimizing FLOPs through global-to-local modeling. They optimized architectures under a fixed FLOPs budget, leading them to favor a model with a six times larger global module compared to the local model.
>
>     In contrast, we studied global-to-local modeling with an emphasis on inference throughput in autoregressive LMs. Specifically, we analyzed *throughput* trends based on the block length and parameter allocation ratio (refer to `Figure 15`) , and our findings reveal that increasing the size of the token decoder (local model) is beneficial for improving inference speed. This interpretation is completely overlooked and under-explored in MEGABYTE, which argue that “*Many of the efficiency advantages of the MEGABYTE design could be realized with the global model alone*”, thus use a significant small local model. This results in a remarkable speedup of up to `20x`, contrasting their reported `1.4x improvement`. New detailed measurements in `Table 1` of the `attached PDF` empirically pinpoints the benefits of the locality aspect, accounting for the `~90% reduction` in walltime at the upper layers compared to vanilla models.
>
>
> 1. Second, as you described, Block Transformers use subword inputs, while MEGABYTE uses byte-level inputs. This enables us to employ an uptraining strategy with initialization techniques that fully leverage existing subword-level language models (refer to `Section 3.7` and `Figure 5a`). Specifically, we demonstrate that with only 10-20% of uptraining, we can almost match the performance of the original model. This is a significant contribution to future research, as it enables the conversion of high-performing LLMs into inference-specialized Block Transformers with minimal additional training cost. Additionally, further exploration of initialization methods could lead to even greater performance improvements and a reduction in the optimal parameter size of Block Transformers.
>
> We would also appreciate acknowledgement for our novel findings and insights regarding global-to-local language modeling, such as the diverse conclusions drawn from analyzing the relationship between block length and parameter allocation ratio in terms of perplexity and throughput (refer to `Section 3.3` and `Section 3.5`), extensive ablation studies on various components, especially token decoder components (refer to `Section 3.4`), IsoFLOP analysis with the inference speed budgets compared to vanilla transformers (refer to `Section 3.6`), or exploring the information contained in context embeddings, which has never been addressed in prior research (refer to `Appendix P` and `Table 5`).
>
> Consequently, we strongly believe that our findings and contributions will provide valuable insights for future research on inference-optimized global-to-local language modeling.

---

> > ### Comment · Reviewer_unsu · 2024-08-08
> >
> > Given that the experiments are indeed extensive, I will increase my score from 3 to 5.

---

> > > ### Author Response · Authors · 2024-08-09
> > >
> > > Thank you for acknowledging our extensive experiments which support the autoregressive inference benefits of our architecture, particularly that of local modeling, which was not fully studied or exploited in previous work.
> > >
> > > Please let us know if you have any further questions or concerns regarding the novelty of our work. We are committed to ensuring our contributions are clearly communicated.

---

### Official Review · Reviewer_Ht4c · 2024-07-13

**Soundness:** 3
**Presentation:** 3
**Contribution:** 3
**Rating:** 6
**Confidence:** 3

**Summary:**

This paper introduces Block Transformer, which is a new architecture that adopts hierarchical global-to-local modeling to autoregressive transformers to mitigate the inference bottlenecks brought by applying self-attention on the global context. In detail, Block Transformer mainly includes three different components: (1) Embedder, which aggregates each block into an input block embedding; (2) Block decoder, which applies self-attention on the full sequence of blocks (rather than tokens) to model global context; (3) Token decoder, which applies self-attention on the sequence of tokens within each block to model local context and decode individual tokens. Evaluation shows that the Block Transformer architecture demonstrates significant gains in inference throughput compared to vanilla transformers with similar perplexity.

**Strengths:**

- The paper explores an important and interesting research direction.
- The improvement on inference throughput achieved by Block Transformer is significant.
- The paper is generally well-written.

**Weaknesses:**

- Block Transformer needs two or three times more parameters than vanilla transformers to achieve similar perplexity.
- It is unclear that, after scaling up vanilla transformers to 7B or 13B level, whether Block Transformer can still achieve similar perplexity with two or three times more parameters.
- More evaluation is required to demonstrate that Block Transformer can effectively leverage full context. While the paper evaluates the perplexity of token positions within a 2K context window to show that Block Transformer can effectively leverage at least 2K tokens of context, experiments on longer contexts that are no shorter than 8K or 16K is also important to show that Block Transformer can indeed effectively leverage global information.

**Questions:**

None beyond the above.

**Limitations:**

The limitation section of this paper is rather comprehensive.

---

> ### Author Rebuttal · Authors · 2024-08-07
>
> We thank you for the thoughtful feedback. We are encouraged that you found our research direction interesting and throughput improvement significant. The weaknesses of our paper are discussed as follows.
>
> .
>
> **W1. More parameters are needed to achieve the similar perplexity.**
>
> KV cache IO and memory size typically impacts generation speed during the decoding process. In standard transformers, larger parameters necessitate larger KV cache sizes, negatively affecting both speed and memory consumption. Conversely, our Block Transformer mitigates this issue by grouping tokens into blocks and performing local modeling on each block embedding (see memory and IO comparison in `Table 1`).
>
> Therefore, despite having two or three times large parameters, the significantly reduced KV cache sizes enable the Block Transformer to demonstrate up to `20 times` faster generation speed, taking into account the total time spent on parameter loading or KV cache loading. Our model also handles much larger batch sizes, making it a more efficient solution for real-world applications. In `Table 1` of the `attached PDF`, we have summarized the actual speed improvements achieved in the attention and FFN operations of both the block and token decoder. Additionally, `Figure 1`, in the `attached PDF`, illustrates which model elements contribute to increased throughput despite having more parameters.
>
> It is also important to note that our Block Transformer is not yet a fully optimized architecture, meaning there is still room for improving performance while maintaining its speed advantage. For example, as confirmed by attention scores (refer to `Appendix O`), incorporating multiple, salient block embeddings in token decoder could substantially enhance performance, potentially reducing the required parameter sizes of Block Transformer. Meanwhile, due to the nature of local modeling, slightly increasing the context length in local modules would have minimal impact on the actual generation time.
>
> Moreover, our token-level modeling structure allows us to uptrain from a well-pretrained checkpoint (refer to `Figure 5a` and `Appendix M`). Further exploration of initialization methods to effectively leverage pretrained models could potentially reduce the parameter size of the Block Transformer to achieve the same perplexity, while significantly reducing the training time as well.
>
> .
>
> **W2. Scaling Models up to 7B or 13B parameters.**
>
> **Performance scaling**: we acknowledge the important of scaling studies. Unfortunately, due to the significant computational resources required for pretraining from scratch, we were unable to verify our findings at larger scale such as 7B or 13B. However, as a proof of concept, we have successfully demonstrated that our proposed modeling approach can achieve similar perplexity across six different scales up to ~1B parameters. Based on our extensive studies, we believe that scaling Block Transformers beyond 7B parameters will still yield compelling perplexity.
>
> **Inference throughput scaling**: independent of perplexity, we compared the throughput of the Vanilla and Block Transformers scaled up to 7B parameters using random initialized weights. We present the maximum throughput (1k tokens/sec) of models not included in `Table 2`, in prefill- and decode-heavy scenarios. We find that our **Block Transformer with 6.9B parameters is still faster than that of a 160M vanilla model**. Assuming that Block Transformer 6.9B performs between that of Vanilla 1.4B and 2.8B, we can still expect *at least* a `6x` speed-up (by comparing with vanilla `1.4B`).
>
> |  | Vanilla 1.4B | Vanilla 2.8B | Vanilla 6.9B | Block 2.8B | Block 6.9B |
> | --- | --- | --- | --- | --- | --- |
> | Prefill Heavy | 0.63 | 0.35 | 0.20 | 7.15 | 4.00 |
> | Decode Heavy | 1.19 | 0.66 | 0.39 | 13.61 | 7.41 |
>
> .
>
> **W3. Lack of experiments on longer contexts like 8K or 16K.**
>
> To further support the effectiveness of our proposed block language modeling in capturing full context, we conducted experiments with an 8K context length (please refer to `Figure 1` in the `attached PDF`). However, due to limited computational resources, we pretrained only a 70M parameter vanilla model and a 170M parameter Block model. Following prior work [1,2,3] that used token position-wise perplexity on the PG19 dataset to demonstrate the utilization of global information in long contexts, we evaluated our Block Transformer in the same manner with 8K context length (refer to `Figure 4b` of the main paper for 2K context window). Even with a extended 8K context window, our models effectively utilized the full context, showing a decreasing loss trend as token position increased, similar to the vanilla model. Besides, consistent with `Table 2` in the main paper, the 170M block model outperformed the 70M vanilla model in terms of perplexity.
>
> It is also worth noting the robustness of our proposed block language modeling in leveraging full context, regardless of the block length. As shown in `Figure 4b` of the main paper, even when varying the block length from 1 to 8, the models exhibited the same loss slope with respect to token positions. This indicates that our block language modeling can capture the full context robustly, regardless of the degree of compression applied to context representations.
>
> .
>
> **Reference**
>
> [1] Yu, Lili, et al. "Megabyte: Predicting million-byte sequences with multiscale transformers."
>
> [2] Xiao, Guangxuan, et al. "Efficient streaming language models with attention sinks."
>
> [3] Zhang, Zhenyu, et al. "H2o: Heavy-hitter oracle for efficient generative inference of large language models."

---

> > ### Comment · Reviewer_Ht4c · 2024-08-13
> >
> > Thanks for the detailed response with clarifications and additional experiments. I believe my concerns of W1 and W3 are mostly addressed, however I still think it is critical to scale models up to at least 7B level for more solid evaluation. So I will be keeping my score the same.

---

> > > ### Author Response · Authors · 2024-08-13
> > >
> > > Thank you for acknowledging our comments.
> > >
> > > We are glad that your concerns regarding parameter requirements and long context capabilities have been addressed.
> > >
> > > We acknowledge the value of 7B parameter experiments, but they were infeasible within our scope. Based on the consistent and significant improvement in throughput from 33M to 1.4B parameters, we believe our work is a solid proof-of-concept for hierarchical global-to-local modeling, demonstrating its significant real-world benefits in subword-level autoregressive inference. This can serve as the foundation for future work, including scaling studies and advanced uptraining schemes (Appendix A), and enable novel research directions which exploit the hierarchical structure, e.g., adaptive computation by dynamically allocating block lengths based on token difficulty.

---

### Author Rebuttal · Authors · 2024-08-07

We extend our gratitude to all the reviewers for providing comprehensive and thoughtful feedback on our manuscript. We appreciate your valuable insights into the strengths and areas for improvement of our work

.

# Core Contributions of Our Work

- **Novelty of approach**: the Block Transformer architecture adopts global-to-local modeling to mitigate key bottlenecks in autoregressive inference which stem from attention.
    - **Difference with pooling transformers**: [1, 2, 3]: our approach incorporates aggregation *and* locality, whereas pooling transformers only use aggregation. The locality of our token decoder is crucial, accounting for `90%` walltime reduction in upper layers, compared to global attention, used in pooling transformers.
    - **Difference with existing byte-level global-to-local models** [4, 5]: Contrary to these works, we find that it is optimal for (a) throughput *and* (b) performance to allocate significant capacity to the token decoder (half of layers), whereas prior work recognize the local model to be just “small” and analogous to a classifier/de-embedder. Our approach achieves `1.5x` throughput compared to MEGABYTE [4], reproduced as a subword level LM.
- **Model efficiency**: our Block Transformers achieve `10—20x gains` in throughput compared to comparable vanilla models on identical hardware. While this comes at the cost of using more parameters, total memory usage is lower than vanilla models, owing to `3--6x` hardware-agnostic reduction in KV cache per sample.
- **Novel architectural contributions**: we propose and compare 3 variants of the embedder and token decoder architecture components. Notably, we design our prefix-based token decoder to exploit the high compute utilization of the token decoder. Loss is reduced by `>0.05` with minimal overhead compared to the local model used in previous work [4].
- **Long-context modeling abilities**: despite limiting global attention to the bottom half of layers, our model is able to utilize contexts up to 2K tokens. We extend this to 8K tokens in `Figure 2 of the attached PDF` and find that our model outperforms vanilla at all token positions. This further supports the viability of a large local component (token decoder).
- **Scalability**: we find that our results scale to models with up to 1.4B parameters. We compare  with baseline vanilla models that achieve equivalent loss and performance on various zero-shot tasks.
- **Uptraining**: we can uptrain vanilla transformers into Block Transformers, using `10% pretraining steps` to approach the performance of training from scratch. This is because we utilize standard transformer components from subword transformers, in contrast to [4, 5]. This significantly lowers development costs and burden to adopt our architecture.

.

# **Summary of Strengths Cited by Reviewers**

- **Impact**: we appreciate reviewers `Ht4c`, `unsu`, `769s` for noting the importance of our research direction, solving the crucial and timely challenge of self-attention bottlenecks in long sequence modeling.
    - We thank reviewer `BaSr` for acknowledging the *bold innovation* of trading parameters for throughput, which enabled our significant throughput gains of `10—20x` under same hardware.
- **Efficiency**: We thank all reviewers for noting our significant improvements in inference efficiency.
- **Experiments**: reviewers `unsu`, `769s`, `BaSr` acknowledged that our experiments are extensive and solid, our modern step (`769s` ) and architecture analysis and ablations (`BaSr`).
- **Writing**: reviewers `Ht4c`, `769s`, `BaSr` acknowledged  that our paper is well-organized and easy to follow.

.

# Additional Material in PDF

- **Figure 1**: **visual recap of the advantages** of coarse global and fine local processing exploited by Block Transformer.
- **Table 1**: detailed measurements on walltime reductions at each inference stage and model component, corresponding to advantages from `Figure 1`.
- **Figure 2**: **Block Transformers can leverage 8K context length**, outperforming its vanilla counterpart at all token positions, while achieving `7—8x` throughput (`Table 2` in the main paper).
- **Figure 3**: Pareto analysis of throughput to language modeling performance using **optimized FlashDecoding [6] kernels. We observe trends identical to our main results** (`Table 2` in the main paper), further supporting the generality of our results.

.

### References

[1] Dai, Zihang, et al. "Funnel-transformer: Filtering out sequential redundancy for efficient language processing."

[2] Nawrot, Piotr, et al. "Hierarchical transformers are more efficient language models."

[3] Clark, Jonathan, et al. “CANINE: Pre-training an Efficient Tokenization-Free Encoder for Language Representation.”

[4] Yu, Lili, et al. "Megabyte: Predicting million-byte sequences with multiscale transformers.”

[5] Mujika, Asier. "Hierarchical attention encoder decoder.”

[6] Dao, Tri, et al. "Flash-Decoding for long-context inference”

---

### Decision · Program_Chairs · 2024-09-25

**Decision:**

Accept (poster)

**Comment:**

The paper proposes a new transformer architecture variant which can alleviate the computational bottlenecks in attention layers of transformer architecture when the sequence length increases. In particular, the proposed architecture uses a block-wise attention on the lower layers which could reduce the complexity by the factor of block size and a local attention on the upper layers which can give fixed cost regardless of the sequence length. The authors claim that combining block-wise attention and local attention gives significant speed-up (10x-20x) while preserving the quality well. The authors emphasize that the local attention is a key contribution to preserve the quality they newly introduced in the paper. The authors provide various experimental results to support the claim.

The reviewers commonly appreciate that the paper is tackling an important problem and showing good efficiency improvements. Also, there's a consensus that the paper is well-written and provide comprehensive evaluations.

There are common questions if the baselines are using flashattention2/flashdecoding which is considered as a common practice to speed-up attention layers. Also, there's a similar question about whether MQA/GQA are applied in the baselines. The authors replied that even with flash-decoding, there exists similar gains, and MQA/GQA is an orthogonal approach which they would explore later. Reviewers also asked to justify the reason the model should use a larger model size to compensate the quality loss from the architecture change. The authors clarified in the rebuttal that the larger model was still faster due to the KV cache reduction. The authors could address most of the issues raised by the reviewers.

Overall, the paper provides a novel solution to an important problem in long sequence generation in LLMs. The paper is well-written and easy to follow. And, the authors provided experimental results to support the claim. One thing to note is reviewer unsu's review quality is low and does not provide much. Considering this and by underweighting reviewer unsu's review, I would recommend the paper to be published in NeurIPS 2024.